# Chemistry and Sensory Characterization of a Bakery Product Prepared with Oils from African Edible Insects

**DOI:** 10.3390/foods9060800

**Published:** 2020-06-18

**Authors:** Xavier Cheseto, Steve B.S. Baleba, Chrysantus M. Tanga, Segenet Kelemu, Baldwyn Torto

**Affiliations:** International Centre of Insect Physiology and Ecology (*icipe*), P.O. Box 30772, Nairobi 00100, Kenya; xcheseto@icipe.org (X.C.); bbaleba@icipe.org (S.B.S.B.); ctanga@icipe.org (C.M.T.); skelemu@icipe.org (S.K.)

**Keywords:** insect oils, fatty acids, flavonoids, vitamin E, bakery product, volatiles, consumer preference

## Abstract

Globally, there is growing interest to integrate insect-derived ingredients into food products. Knowledge of consumer perception to these food products is growing rapidly in the literature, but similar knowledge on the use of oils from African edible insects remains to be established. In this study, we (1) compared the chemistry of the oils from two commonly consumed grasshoppers, the desert locust *Schistocerca gregaria* and the African bush-cricket *Ruspolia differens* with those of olive and sesame oils; (2) compared the proximate composition of a baked product (cookie) prepared from the oils; (3) identified the potential volatiles and fatty acids contributing to the aroma and taste; and (4) examined acceptance and willingness to pay (WTP) for the baked product among consumers with no previous experience of entomophagy. Our results showed that the insect oils were compositionally richer in omega-3 fatty acids, flavonoids, and vitamin E than the plant oils. Proximate analysis and volatile chemistry revealed that differences in aroma and taste of the cookies were associated with their sources of oils. Consumers’ acceptance was high for cookies prepared with *R. differens* (95%) and sesame (89%) oils compared to those with olive and *S. gregaria* oils. Notably, cookies prepared with insect oils had more than 50% dislike in aroma and taste. Consumers’ willingness to pay for cookies prepared with insect oils was 6–8 times higher than for cookies containing olive oil, but 3–4 times lower than cookies containing sesame oil. Our findings show that integrating edible insect oils into cookies, entices people to ‘‘take the first step” in entomophagy by decreasing insect-based food products neophobia, thereby, contributing to consumers’ acceptance of the baked products. However, future research should explore the use of refined or flavored insect oils for bakery products to reduce off-flavors that might have been perceived in the formulated food products

## 1. Introduction

A recent report by the United Nations estimates the increase in the world’s population to reach 9.7 billion in 2050 from the current 7.7 billion, with most increases occurring in developing countries, especially sub-Saharan Africa, where the population is expected to double by 2050, representing a 99% increase [1]. These increases are expected to impact the demand for food and protein sources [2,3], which will put pressure on the traditional sources of food including plants, livestock, and fish [4]. As such, interest has shifted to ‘novel food products’ to secure a nutritious and sustainable diet for the growing human population. Insects represent promising and sustainable alternatives as proposed by various agencies including the Food and Agricultural Organization (FAO) for their inclusion into the food and feed value chains [5,6,7]. Several studies have shown that insects are excellent sources of protein, fats, oils, and other nutrients for improved nutrition and health of humans and animals [4,5,6,8,9].

In sub-Saharan Africa (SSA), over 500 insect species are widely consumed, most of which are harvested seasonally from the wild [7,10,11]. At the International Centre of Insect Physiology and Ecology (*icipe*), we have successfully developed technologies for mass production of several species of edible insects and demonstrated their technical and economic feasibility as protein-based ingredients for food and feeds [12,13]. Similar studies have been carried out by researchers in various parts of the world, leading to the production and marketing of insect-based products in various forms: cake, paste, powder, and oils [14,15]. The production of insects as food and feed not only translates into a better ecological footprint, but it also significantly lowers greenhouse gas emissions, as well as water and land requirements compared to beef and other livestock [4,5].

Among the edible insect ingredients, oils are recognized worldwide for their nutritional value, health benefits, and are also appreciated for their aroma and taste [16]. As such, their production contributes immensely to the economies of many countries. According to a report by Persistence Market Research, the global edible insect oil market is expected to increase from $83.4 billion to $130.3 billion by 2024, whereas the global omega-3 fatty acid market size is projected to reach $3.77 billion by 2025, registering a compound annual growth rate of 7.4% [17]. Additionally, according to a report by Grand View Research Inc. the global skin care products market size will be $183 billion by 2025 [18]. Therefore, in recent years, there has been an emerging new research dimension on edible insect oils as sources of compounds of dietary and therapeutic value. Previous research has reported desert locust *Schistocerca gregaria* and African bush-cricket *Ruspolia differens* to be rich in fats and oils (12% and 48% dry weight, respectively). This suggests that they can potentially be exploited for their saturated and unsaturated fatty acids [19,20]. Other unconventional sources of oils have been harnessed from several insects using different solvents and their composition determined [21,22,23]. For instance, an analysis of the oil obtained from a hexane or dichloromethane extract of the larvae of the house fly *Musca domestica* Linnaeus, identified several essential fatty acids [21]. Additionally, the physico-chemical composition (thermal behavior, color, and aroma compounds) of an aqueous-based oil extract of four commercially-produced insects in Europe (the yellow mealworm, lesser mealworm, house cricket, and Dubia cockroach) has been reported [22]. Recently, black soldier fly larvae fat has also been used as a butter substitute up to 25% in cakes and cookies, while in waffles, the substitution could go up to 50% without influencing consumers’ acceptance [23]. 

The quality of insect oil is dependent on the extraction technique and insect species. In this study, we used the aqueous-based extraction method to obtain oils from two commonly consumed grasshopper species in Africa (the desert locust *Schistocerca gregaria* and the African bush-cricket *Ruspolia differens*) to (1) compare the chemistry of the oils with those of two commercially-produced plant (olive (*Olea europaea*) and sesame (*Sesamum indicum*)) oils, (2) determine and compare the proximate composition of a baked product (cookie) prepared from these oils, (3) identify the potential volatiles and fatty acids contributing to the aroma and taste of the baked product, and (4) examine acceptance and willingness to pay (WTP) for the baked product.

## 2. Materials and Methods 

### 2.1. Insects

The desert locust *S. gregaria* and the African bush-cricket *R. differens* (Order: Orthoptera), were selected for this study because they are widely consumed in several countries in sub-Saharan Africa [7]. They were mass reared at the Animal Rearing and Containment Unit (ARCU) of the International Centre of Insect Physiology and Ecology (*icipe*), Nairobi, Kenya (1.2219° S, 36.8967° E; 1600 m ASL) and fed on a diet consisting of wheat seedling, maize leaves, and wheat bran. At maturity, both the desert locust and the African bush-cricket were harvested and processed for oil extraction as described in Section 2.2. The colonies of desert locust and the African bush-cricket were maintained at 30 ± 2 °C, relative humidity (RH) of 65 ± 5% and a photoperiod of 12:12 light–dark cycle. A digital thermo-hygrometer TH-812E was maintained inside each of the rearing rooms to track changes in temperature and RH. 

### 2.2. Extraction of Insect Oils 

Ten kilograms (10 kg wet weight each) of adult *R. differens* and *S. gregaria* were extracted following the procedure previously described in aqueous media [24] with some modifications. The insects were first killed by freezing them at −80 °C for 3 h. The dead insects were cut into small pieces, transferred into a conical flask containing distilled deionized water (5 *v*/*w*), and then heated at 80 °C for 3 h, with gentle stirring using a magnetic stirrer. To get rid of the solids, the mixture was filtered and gently squeezed through a two-layer gauze cloth and the filtrate collected into a separating funnel and left overnight to separate. The upper layer was centrifuged for 30 min at 14,000 rpm, and the supernatant (insect oil) collected, weighed, and stored at −20 °C for further analysis. The average oil yield was 8% and 4% for *R. differens* and *S. gregaria*, respectively. The insect oils were extracted in three replicates for each species using a new group of insects. 

### 2.3. Plant Oil

Sesame extra virgin cold pressed and unrefined oil (Nakuru, Kenya) and Fragata extra virgin olive oil (Angel Camacho Alimentacion S.L, Spain; 500 mL each) were purchased from a local supermarket in Nairobi, Kenya. 

### 2.4. Gas Chromatography Coupled to Mass Spectrometry (GC-MS) Analysis of Fatty Acids

The fatty acid (FA) compositions of the insect and plant oils and baked products (cookies) made from the respective oils (20 mg each), were analyzed as fatty acid methyl esters (FAMEs) following previous methods [25]. A solution of sodium methoxide (15 mg/mL) was prepared in dry methanol and was added (500 µL) to the samples. The samples were vortexed for 1 min, sonicated for 5 min, and incubated at 60 °C for 1 h, thereafter quenched by adding 100 μL deionized water followed by vortexing for another 1 min. The resulting methyl esters were extracted using gas chromatography (GC)-grade hexane (1000 µL; Sigma–Aldrich, St. Louis, MO, USA) and centrifuged at 14,000 rpm for 5 min. The supernatant was dried over anhydrous Na_2_SO_4_ and analyzed (1.0 µL) by GC-MS on a 7890A gas chromatograph linked to a 5975 C mass selective detector (Agilent Technologies Inc., Santa Clara, CA, USA). The GC was fitted with a (5%-phenyl)-methylpolysiloxane (HP5 MS) low bleed capillary column (30 m × 0.25 mm i.d., 0.25 µm; J&W, Folsom, CA, USA). Helium at a flow rate of 1.25 mL min^−1^ served as the carrier gas. The oven temperature was programmed from 35 °C to 285 °C, with the initial temperature maintained for 5 min, with a rise at 10 °C min^−1^ to 280 °C, and then held at this temperature for 20.4 min. The mass selective detector was maintained at ion source temperature of 230 °C and a quadrupole temperature of 180 °C. Electron impact (EI) mass spectra were obtained at the acceleration energy of 70 eV. Fragment ions were analyzed over 40–550 m/z mass range in the full scan mode. The filament delay time was set at 3.3 min. Serial dilutions of the authentic standard methyl octadecenoate (0.2–125 ng/μL) and hexanal (1–280 ng/μL) were analyzed by GC-MS in full scan mode to generate a linear calibration curves (peak area vs. concentration) which gave coefficient of determinations R^2^ = 0.9997) and 0.9997 for methyl octadecenoate and hexanal. These regression equations were used for the external quantification of the different fatty acids and volatile organic compounds (VOCs) respectively. 

A **Hewlett-Packard** (HP Z220 SFF intel xeon) workstation equipped with ChemStation B.02.02. acquisition software was used. The mass spectrum was generated for each peak using Chemstation integrator set as follows: initial threshold = 5, initial peak width = 0.1, initial area reject = 1, and shoulder detection = on. The compounds were identified by comparison of mass spectral data and retention times with those of authentic standards and reference spectra published by library–MS databases: National Institute of Standards and Technology (NIST) 05, 08, and 11. The insect and plant oils as well as the insect-based oil cookies were analyzed for FAMEs in triplicate, with each replicate collected from a different batch of respective samples.

### 2.5. GC-MS Analysis of Vitamin E

Each insect and plant oil (300 mg), was transferred into a 10 mL glass vial containing a mixture of hexane, methanol and distilled deionized water (2:1:2, 5 mL), vortexed for 30 s, sonicated for 30 min and centrifuged at 14,000 rpm for 5 min. The supernatant was dried over anhydrous Na_2_SO_4_, evaporated to dryness under a gentle stream of N_2(g)_ before derivatizing any residual fatty acids following the protocol in the fatty acid section to limit the matrix interference, analyzed and quantified by GC-MS as described for fatty acids. Three replicates were carried out, with each replicate done on a different batch of oil.

### 2.6. Liquid Chromatography Tandem Mass Spectrometry (LC-MS/MS) Analysis of Flavonoids

An oil sample from the insect and plant (300 mg) was transferred into a 2 mL Eppendorf tube containing LC-MS-grade acetonitrile (1000 µL; Sigma–Aldrich, St. Louis, MO, USA), vortexed for 1 min, sonicated for 30 min and centrifuged at 14,000 rpm for 5 min. The acetonitrile soluble supernatant was carefully transferred into an autosampler vial after which 0.2 μL was analyzed on a Waters Xevo TQ-S LC-MS/MS (Waters Corp., Milford, MA). The chromatographic separation was done on a Waters ACQUITY ultra-performance liquid chromatography (UPLC) I-class system fitted with an ACE C18 column (250 mm × 4.6 mm i.d., 5 µm) from Advance Chromatography Technologies, Aberdeen, Scotland. The mobile phases used were water (A) and acetonitrile (B), each with 0.01% formic acid. The following gradient was used: 0 min, 5% B; 0–2 min, 5–20% B; 2–4 min, 20% B; 7–8.5 min, 20–60% B; 8.5–10 min, 60% B; 10–15.0 min, 60–80% B, 15–19 min, 80% B; 19–20.5 min, 80–100% B; 20.5–23 min, 100% B; 23–24 min, 100–5% B; and 24–26 min, 5%. The flow rate was held constant at 0.4 mL min^−1^. 

The Ultra Performance Liquid Chromatography (UPLC) system was coupled to a Xevo TQ-S LC-MS/MS (Waters Corp., Milford, MA, USA) equipped with an electrospray ionization source operated in full scan positive mode. Data were acquired from a mass range of m/z 100 to 1500. Other parameters were: scan time, 1 s; capillary voltage, 3.3 kV; sampling cone voltage, 40 V; source offset, 40 V; source temperature, 150 °C; desolvation gas temperature, 250 °C; nitrogen desolvation flow rate, 600 L/h; cone gas flow rate,150 L/h; and collision gas Argon at 3.5 X 10^−3^ mbar.

Data acquisition was achieved with MassLynx version 4.1 SCN 712 (Waters). The mass spectrum was generated for every peak and potential assignments done using online literature data METLIN, (http://metlin.scripps.edu); ChemSpider, (http://www.chemspider.com); ChemCalc (https://www.chemcalc.org), and a commercial National Institute of Standards and Technology tandem mass spectral library (NIST MS/MS) and other published literature [26,27,28]. The identities of quercetin, luteolin, kaempferol, rutin, and apigenin in the samples were confirmed with commercially purchased samples by co-injections. Quercetin (≥95%), luteolin (≥98%), kaempferol (≥90%), rutin hydrate (≥94%), and apigenin (≥95%) were purchased from Sigma–Aldrich (St Louis, MO). Serial dilutions of the flavonoid, rutin (Rutin hydrate ≥ 94%; 1–100 ng/µL) were also analyzed by LC-MS/MS to generate a linear calibration curve (peak area vs. concentration) with the following equation; (y = 3137x − 1353.1 (R^2^ = 0.9993)), which was used for external quantification of all the flavonoids. Three replicates were carried out with each replicate done on a different oil batch.

### 2.7. Analysis of Headspace Volatiles

Headspace volatiles were collected from the insect and plant oils and cookies prepared from the respective oils. The oil (3 g) was placed in a 13 mL autosampler glass vial (Shimadzu, Kyoto, Japan), while the cookies (30 g) were crushed into pieces before being transferred into a 250 mL quick fit chamber Agricultural Research Service (ARS), Gainesville, FL, USA). Activated charcoal-filtered and humidified air was passed over the samples using a push-pull Gast pump (Gast Manufacturing Inc., Benton Harbor, MI, USA) at a flow rate 340 mL/min. The volatiles were trapped on to pre-cleaned with GC-grade dichloromethane (DCM) Super-Q traps (30 mg, Analytical Research System, Gainesville, FL, USA) at a flow rate of 170 mL/min using a Vacuubrand CVC2 vacuum pump (Vacuubrand, Wertheim, Germany) for 12 h. The Super-Q traps were then eluted with 200 μL DCM into 2 mL clear glass vials, each containing 250 μL conical point glass inserts (Supelco, Bellefonte, PA, USA) and immediately analyzed using GC-MS. All volatiles were trapped in triplicates using different batches of oils and cookies.

### 2.8. Wheat Flour Blended Cookies Baked with Insect and Plant Oils 

The local recipes used to prepared the dough for the cookies consisted of 1300 g of flour, 500 g of sugar, 300 mL of each oils (Treatment–T1 (sesame oil), T2 (African bush-cricket oil), T3 (olive oil) and T4 (desert locust oil)), 5 g of baking powder, 40 g of eggs, and 150 mL of milk to prepare the required consistency of the dough. The firm dough was rolled out to 5 mm thickness in a baking tray and then cut into 5 cm diameter circles with 5 mm thickness with a cookie cutter. The cut dough samples were then placed on a non-sticky aluminum tray and baked in a pre-heated oven (BISTROT 665; BestFor^®^, Ferrara, Italy) at 200 °C for 10 min. A total of 4250 cookies (850 cookies/treatment) were prepared. Baked cookies were assessed for their nutritional and organoleptic qualities.

### 2.9. Proximate Analysis of Wheat Flour Cookies Baked with Insect and Plant Oils

The moisture content was determined by drying the cookies in an oven at 105 °C for 24 h [29]. The nitrogen content was determined using the Kjeldahl method [29] and later converted to crude protein content using a nitrogen factor for the crude protein calculation of 6.25 [30]. The ash content was determined by drying the sample in a furnace at 550 °C until the weight remained constant [29]. Total carbohydrate was calculated by difference using standard methods [29]. All parameters discussed above were determined in quadruplicate per sample and expressed as a percentage. 

### 2.10. Sensory Analysis

The sensory attributes of the wheat-oil cookies in relation to color, aroma, taste, overall acceptance, and willingness to buy were evaluated. Ranking test was used to evaluate the perceptible differences in intensity of an attribute among samples using a 3-point hedonic scale (where, 1 represented dislike, 2 neutral, and 3 like), with equivalent intervals between the categories [31]. The sensory evaluation panel comprised of mixed semi-trained 103 insect-based food neophobia panelists randomly selected from an institution of higher learning in Kenya. The members of the trained panel [32] were selected based on their experiences in descriptive analysis of different food products and familiarity with the sensory quality of cookies, though without any prior experience in insect consumption. The assessments were carried out in a sensory laboratory room that almost fulfilled the requirements of the International Organization for Standardization (ISO) standard [33], in individual booths equipped with pen and questionnaires for data collection and processing. Together with the samples, the panelists received a cup of room temperature spring water for cleaning their mouths. Panelists were asked to focus first on the aroma and color, and next on the taste of the separated cookies. The panelists, together with the panel leader, established the descriptions of the main sensory attributes of cookies using a standard procedure [32].

### 2.11. Data Analysis

All the analyses were conducted in R software version 3.5.0 [34]. Data from oil yield were normally distributed (Shapiro–Wilk test: *p* > 0.05) and their variance were similar (Bartlett’s test: *p* > 0.05), therefore, we used the unpaired *t*-test to compare the oil yield between *R. differens* and *S. gregaria*. We performed principal component analysis (PCA) to assess the difference in saturated fatty acid (SFA), monounsaturated fatty acid (MUFA), and polyunsaturated fatty acid (PUFA) among the different oil types. To visualize the relative abundance of the different classes of fatty acids (SFAs, MUFAs, and PUFAs) and their variation across the different insects and plant oils, we generated 100% stacked bars using excel and compared their proportions using a chi-square test. We used the same statistical test to compare the different classes of FAME in cookies prepared from the insect and plant oils. We employed the test of analysis of variance (ANOVA) followed by the Student–Neuman–Keuls (SNK) post-hoc multiple comparisons test to compare the concentration of the omega-3 acids across the different oil and cookie types. We performed the one-way analysis of similarities (ANOSIM) using Bray–Curtis dissimilarity matrix to compare the chemical profiles of different oil volatiles. Based on the similarity percentages (SIMPER) analysis, we determined the relative contribution of different compounds to the dissimilarity among volatiles from the different oils. We visualized this using the non-metric multidimensional scaling approach. For proximate analysis, data was subjected to analysis of variance to examine the effect of oil type on moisture, ash, fiber, carbohydrate, and crude protein, respectively. Means were separated using Student–Neuman–Keul’s test. For sensory analysis, a quantitative descriptive analysis was performed to determine the sensory characteristics of the samples. Sensory evaluation scores data on color, aroma, taste, and overall acceptance were summarized as percentage on each score category. All statistical results were considered significant when *p* < 0.05.

## 3. Results

### 3.1. Yields of Insect Oils 

There was a significant difference in oil yield between *R. differens* and *S. gregaria* (t = −4.58, df = 3.76, *p* = 0.012). *R. differens* yielded twice as much more oil than *S. gregaria* (Figure 1). 

### 3.2. Chemical Profiles of Insect and Plant Oils and Cookies 

The profiles of the insect and plant oils revealed fatty acids, flavonoids, and vitamin E (Table 1 and Table 2, Figure 2 and Figure 3).

Of the total 73 fatty acids detected in the insect and plant oils, saturated fatty acids (SFAs) contributed 35 components (approximately 48%), monounsaturated fatty acids (MUFAs) 22 (30%), and polyunsaturated fatty acids (PUFAs) 16 (22%), distributed as follows: SFAs *S. gregaria* (25), *R. differens* (25), sesame (17), olive (15); MUFAs *S. gregaria* (11), *R. differens* (11), sesame (10), olive (9); and PUFAs *S. gregaria* (14), *R. differens* (13), sesame (5), and olive (4; Table 1). Although the pattern of the composition of the fatty acids was similar for both the insect and plant oils, there were some qualitative differences attributed to six SFAs (C_11_–C_13_), detected only in the insect oils, and eight SFAs (C_22_–C_28_), detected only in the plant oils. The MUFA composition also varied with the oil. For instance, the three MUFAs C_6_–C_14_ were detected only in the insect oils, whereas the three C_10_–C_14_ MUFAs were detected only in the plant oils. Additionally, the three PUFAs C_20_–C_22_ were detected only in the insect oils. Notably, five SFA methyl esters (methyl dodecanoate, methyl tetradecanoate, methyl pentadecanoate, methyl octadecenoate, methyl nonadecanoate), two MUFA methyl esters (methyl (10*Z*)-heptadecenoate, methyl (9*Z*)-octadecenoate) and three PUFA methyl esters (methyl (9*Z*,12*Z*)-octadecadienoate, methyl (9*Z*,12*Z*,15*Z*)-octadecatrienoate, methyl (9*Z*,11*E*,13*E*)-octadecatrienoate) were consistently detected across all the samples analyzed.

The fatty acid profile of the cookies mirrored that of their parent oils distributed as follows: SFAs *S. gregaria* (21), *R. differens* (16), sesame (17), olive (11); MUFAs *S. gregaria* (10), *R. differens* (8), sesame (10), olive (4); and PUFAs *S. gregaria* (8), *R. differens* (8), sesame (5), olive (4; Table 1). Compositionally, we detected 1.7-fold more fatty acid methyl esters (FAMEs) in the insect oils than in the plant oils. Additionally, we detected approx. 1.6-fold more FAMEs in the insect and plant oils than in their respective cookies. These patterns are shown in the biplots from the principal component analysis (PCA). The PCA analysis clustered together all the insect oils and cookies prepared with insect oils into two dimensions, explained as 90.9%, 98.9%, and 98.9% of the total variation in the SFAs (Figure 4A), MUFAs (Figure 4B), and PUFAs (Figure 4C), respectively. 

For SFA, PC1 explained 68.2% of total variation contributed by methyl 13-methyltetradecanoate, methyl 3-methylpentadecanoate, and methyl 10-methylundecanoate, while PC2 explained 22.7% of total variation contributed by methyl 14-methylhexadecanoate, methyl nonadecanoate, and methyl heptadecanoate. For MUFA, PC1 explained 83.4% of total variation contributed by methyl 8-heptadecenoate, methyl (*E*)-6-octadecenoate and methyl (*Z*)-6-octadecenoate, while PC2 explained 15.5% variation contributed by methyl (Z)-11-eicosenoate, methyl (Z)-15-tetracosenoate, and methyl (Z)-11-docosenoate. For PUFA, the most influential constituents were methyl (3R,6*E*,10*E*)-3,7,11,15-tetramethylhexadeca-6,10,14-trienoate, methyl (5*Z*,8*Z*,11*Z*,14*Z*)-eicosatetraenoate, and methyl (11*E*,14*E*)-eicosadienoate accounting for PC1 value of 83.3%, while PC2 explained 15.5% of total variation contributed by methyl (9*Z*,11*E*,13*E*)-octadecatrienoate, methyl (10*E*,12*Z*)-octadecadienoate and methyl (5,12)-octadecadienoate.

Additionally, the 100% stacked bars qualitatively and quantitatively showed the variation of FAMEs across the different oil categories (Figure 5).

The data indicates that all the insect oils and cookies made from the respective insect oils had similar amounts (%) of SFAs, MUFAs, and PUFAs. The pattern was different for the plant oils and their respective cookies where the highest proportion of MUFAs was found in olive oil and its cookies (64% and 50% respectively). For sesame oil, PUFAs (46%) was highest in sesame oil and SFAs (41%) was highest in its cookies.

The most influential constituents contributing to the highest concentration of the total (%) FAMEs for each category (SFA, MUFA, and PUFA) in the oils of insects, plants and cookies is shown in Table 1. Of the 35 SFAs identified, methyl hexadecanoate contributed the highest concentration across all the oils and cookies, except for the oil of *R. differens*, where it was the second most abundant concentration following methyl 10-hexadecanoate. 

In addition to quantitative differences observed in the MUFAs, the methyl esters methyl 8-heptadecenoate, methyl (6*Z*)-octadecenoate, methyl (6*E*)-octadecenoate, methyl 16-octadecenoate and methyl (11*E*)-eicosenoate were only detected in the insect oils, whereas the concentration of the MUFA methyl (9*Z*)-octadecenoate (oleic acid) was relatively higher across all the oils and cookies.

Of the 16 PUFAs identified, methyl (9*Z*, 12*Z*)-octadecadienoate (linoleic acid, LA) contributed the highest proportion in the oils of *S. gregaria* (38%), *R. differens* (42%), sesame (98%), and olive (43%). For the cookies, (methyl (10*E*,12*Z*)-octadecadienoate contributed the highest proportion in the cookies prepared from the oils of *S. gregaria* (47%) and *R. differens* (47%), while methyl (9*Z*,12*Z*)-octadecadienoate was the most dominant PUFA in the cookies prepared from sesame (98%) and olive (63%) oils. The amounts of LA detected in the insect and sesame oils were 1.5- to 3.8-fold higher than found in their respective cookies. On the other hand, the LA level in the cookies made from olive oil was 2.8-fold higher than that found in the parent oil.

The concentrations of omega-3 fatty acids, namely α-linolenic acid (ALA; Figure 6Ai; F_(3,8)_ = 205.1, *p* < 0.001), eicosapentaenoic acid (EPA; Figure 6Aii; F_(3,8)_ = 992.6, *p* < 0.001) and docosapentaenoic acid (DHA; Figure 6Aiii; F_(3,8)_ = 61.17, *p* < 0.001) varied significantly across the insect and plant oils. Also, in the baked products, the concentrations of ALA (Figure 6Bi; F_(3,8)_ = 17.4, *p* < 0.001), EPA (Figure 6Bii; F_(3,8)_ = 23.01, *p* < 0.001), and DHA (Figure 6Biii; F_(3,8)_ = 9.36, *p* < 0.001) varied significantly between cookies baked with the plant and insect oils. The concentrations of ALA in the oils were 1.3–2.5 times higher than in their corresponding baked products. 

Beside the essential fatty acids described above, ω-6/ω-3 ratios were found to vary considerably among the different oils and their associated baked products (χ^2^ = 57.6, *p* < 0.001) and ranged between 1.4 to 12.5 (Table 1). Notably, the ω-6/ω-3 ratios of the insect and olive oils were <5 but these values increased between 1.6–7.7-fold in the cookies.

Additional chemical analysis of the insect and plant oils detected vitamin E and specific flavonoids. The concentration of vitamin E detected in the oils significantly varied across the different oil sources (F_(3,8)_ = 11.04, *p* < 0.05). Compared to olive and sesame oils, the vitamin E contents in the *S. gregaria*, *R. differens* oils were 2.8–6-fold higher than the plant oils (Figure 2). Vitamin E levels in the cookies were not assessed.

We identified 8 flavonoids (Table 2), of which, 4 (50%) each were identified in the oils of the edible insects *R. differens* and *S. gregaria*, compared to the plant sesame (3) and olive (2) oils. The flavonoids detected included quercetin (F_(3,8)_ = 14.38, *p* < 0.0001), luteolin (F_(3,8)_ = 380.5, *p* < 0.0001), kaempferol (F_(3,8)_ = 5.96, *p* < 0.001), orientin (F_(3,8)_ = 4.17, *p* < 0.05), hesperidin (F_(3,8)_ = 4.19, *p* < 0.05), sesamoside (F_(3,8)_ = 333.9, *p* < 0.001), rutin (F_(3,8)_ = 1.80, *p* < 0.05), and apeginin (F_(3,8)_ = 27.12, *p* < 0.001). Quercetin, luteolin, kaempferol, and orientin (Figure 3A–D) were identified in the two insect oils, whereas sesame oil was rich in hesperidin and sesamoside (Figure 3E,F), with rutin and apeginin identified in olive oil (Figure 3G,H). Flavonoid levels were not assessed in the cookies.

Chemical profiles of the aroma of the insect and plant oils and their respective cookies revealed a total of 73 volatile organic compounds (VOCs; Table 3). The VOCs were predominantly carboxylic acids, aldehydes, and alcohols and their concentrations varied significantly across the different sources (one-way ANOSIM based on Bray–Curtis dissimilarity, R = 0.974, *p* < 0.0001). The non-metric multidimensional scaling plot (Figure 7A) with a stress value of 0.16 (great representation of dissimilarities), clustered insect and plant oils and their respective cookies well (Figure 7B). The difference in volatile compositions between the oils and cookies can be attributed to 3-methylbutanoic acid (6.3%), 2,3-butanediol (3%), and 2-methylbutanoic acid (2.9%) in the insect oils. Contrastingly, (*Z*)-3-hexenyl butanoate (7.1%) and limonene (3.5%) were associated with the plant olive and sesame oils.

### 3.3. Nutritional Composition of Cookies Prepared with Insect and Plant Oils

Proximate analyses of the cookies baked with insect and plant oils are presented in Table 4. The moisture content (F_(3,12)_ = 19.96; *p* < 0.0001), ash (F_(3,12)_ = 46.67; *p* < 0.0001), fiber (F_(3,12)_ = 807.1; *p* < 0.0001), crude protein (F_(3,12)_ = 22.7; *p* < 0.0001), and carbohydrate (F_(3,12)_ = 1098; *p* < 0.0001) values of the cookies varied significantly across the various treatments (F_(3,12)_ = 19.96; *p* < 0.0001). Cookies baked with *S. gregaria* oil had a significantly higher crude protein content than the other baked cookies.

### 3.4. Sensory Evaluation and Consumer Acceptability

The color, appearance, and sensory attributes of wheat-insect oil composite cookies are presented in Figure 8 and Figure 9. 

More than 80% of panelist preferred the color of cookies baked with sesame, olive and *R. differens* oils (Figure 9). For aroma and taste, <20% of the respondents recommended cookies baked with the insect oils. However, panelists’ acceptance of cookies baked with the oils from *R. differens* and sesame oils were highest at 89% and 95%, respectively (Figure 9). There was a low positive correlation (*r* = 0.4088; t = 9.0704; df = 410; *p* < 0.0001) between color and overall acceptance of the baked cookies. Contrastingly, there was a significantly positive correlation between taste (*r* = 0.8372; t = 30.998; df = 410; *p* < 0.0001) and aroma (*r* = 0.7847; t = 25.629; df = 410; *p* < 0.0001) among the panelists. The willingness to pay (WTP) for cookies baked with sesame oil was 89%, which was approximately 3–23-fold higher than the WTP for cookies baked with the oils of *R. differens* (30%), *S. gregaria* (22%), and olive oils (4%) (Figure 10). 

## 4. Discussion

### 4.1. Chemistry, Nutritional, and Health Properties of Oils Derived from Insect and Plant as Well as Their Respective Bakery Product

Our results showed that of the two edible insects, the African bush-cricket *R. differens* is a better source of oil in terms of both quality and quantity than the desert locust. This differential oil yield between the two grasshopper species might be associated with dietary-dependent feed requirements based on their conversion efficiencies and general lipid needs [12], developmental stage, sex, and environmental temperature of the insects and method of oil extraction [14]. Additional research using many species from the order Orthoptera is needed to confirm these findings. The results also suggest that the presence of fatty acids, higher levels of flavonoids, and vitamin E in the insect than plant oils could serve as potential suitable biomarkers for their nutritional qualities for use as food ingredients. Interestingly, vitamin E is known to play an important role as an antioxidant in reducing the number of radicals in the human body. It also helps in preventing lipid oxidation, maintenance of skin health, and supports the immune system and cell function [35]. In insects, vitamin E plays a role in reproduction and also as an anti-oxidant [36]. Likewise, flavonoids, whose sources for human health are mainly fruits and vegetables [37], also play an important role as anti-oxidants, and other benefits including anti-inflammatory and anti-microbial effects [38]. Previous research has also reported antioxidant activities in water-soluble extracts of grasshoppers, silkworms, and crickets [39]. Although the current study is not exhaustive, it provides evidence that insect oils may contain additional antioxidants and nutritional components, which would require future research to allow for their full exploitation in human food and animal feed. 

Chemical analysis showed that the general pattern of fatty acids in the oils and their respective cookies remained the same although there were variations in their concentrations, especially in the cookies. Notably, the dominant saturated fatty acids in both the insect and plant oils and their respective cookies remained methyl hexadecanoate and its isomer methyl 10-hexadecanoate This suggests that baking did not influence the general pattern of the composition of the fatty acids. Interestingly, the fatty acid values for SFAs, MUFAs, and PUFAs found in the current study compared favorably with the values reported by other researchers [16,40,41,42], which confirmed the sensitivities of our extraction and analysis of the oils. For example, the SFA, and combined MUFA and PUFA levels in *S. gregaria* oil were 50.4% and 49.5%, respectively, which agrees with previously reported levels of 43.6% and 53.5% [40,43]. The SFA, MUFA, and PUFA levels in the oil of *R. differens* were 45%, 20%, and 34% respectively, which agrees with the previously reported values of SFA 31–35%, MUFA 21–30%, and PUFA 33.7–44% [44,45]. In olive oil, we found a SFA level of 34%, MUFA 64%, and PUFA 2% compared to previously reported levels of SFA 5–35%, MUFA 55–83%, and PUFA 3–12% [41]. In sesame oil, we found a PUFA level of 64% compared to 42–54% reported previously for the same oil [42]. 

The SFAs, MUFAs, and PUFAs detected in the various oils and cookies are major sources of energy and for certain physiological functions in living organisms. For instance, in insects, the SFA hexadecanoic acid and octadecanoic acid are known to play a role in defense against pathogens [46], while the presence of dodecanoic acid, hexadecanoic acid, octadecanoic acid, and tetradecanoic acid in foods is associated with flavors. Additionally, these fatty acids are important reagents for making soap [47]. In the present study, oleic acid was identified as the most abundant MUFA in all the treatments. Oleic acid has previously been reported as the most abundant fatty acid in both insects [40] and humans [48]. It is important to note that there are important benefits associated with the presence of oleic acid in living organisms, for example, in insects, it regulates the fluidity of cell membranes by serving as a major precursor for the biosynthesis of waxes and semiochemicals [49], while in humans, it serves as a cholesterol-lowering agent associated with heart diseases [49] and as an anticancer and anti-inflammatory agent [50]. Additionally, it plays a crucial role during pregnancy-lactation [51], promotes cell viability [52], and serves as a source of energy and a precursor for the biosynthesis of linoleic acid required for production of arachidonic acid involved in cell regulation [48]. Thus, the detection of oleic acid in all the oils and bakery products (cookies) suggests that it may contribute to their overall nutritional quality.

Of the omega-6 fatty acids, it is important to note that the concentrations of linoleic acid (LA) recorded in all the oils and their corresponding bakery products were higher than the levels reported in conventional meat products, such as fish, beef, and chicken [53]. Linoleic acid is widely known for its role in the synthesis of arachidonic acid a precursor for various hormones, such as prostaglandins, thromboxanes, and leukotrienes used for the regulations of many physiological process [54]. Linoleic acid (LA) and omega-3 fatty acids (α-linolenic acid (*Z*,*Z*,*Z*)-9,12,15-octadecatrienoate (ALA)) are essential fatty acids in humans, obtained predominantly from dietary sources. The variation in the concentration of linoleic acid in the different cookies could be partly attributed to the preparation of the dough which may not be completely uniform across the different treatments and its response to the high temperature used in the baking process.

Given that we detected higher concentrations of the omega-3 fatty acids eicosapentaenoic acid (EPA) and docosapentaenoic acid (DHA) in both the insect oils and their associated cookies than the plant oils and their cookies, reveals that both oils retain their nutritional benefits when used in food products. Omega-3 fatty acids are known for the range of health benefits confer to humans [55,56]. For example, previous studies have demonstrated that eicosapentaenoic acid (EPA) plays a major role in the management of diseases such as high blood pressure, schizophrenia, cystic fibrosis, and Alzheimer’s disease [55]. Furthermore, docosapentaenoic acid (DHA) has been widely used as a supplement in baby formulas to promote mental development, and in the treatment of type 2 diabetic cases and other diseases [56]. Health benefits associated with the combined use of EPA and DHA include the treatment of skin infections and Crohn’s disease [55]. 

Interestingly, the ω-6/ω-3 ratios of all the insect and olive oils found in the current study fall within the anticipated range of the required daily intake (<5:1) recommended by World Health Organization (WHO). On the other hand, they were high (>5:1) in the cookies prepared with the insect and sesame oils. These differences could be associated with the high temperatures used to bake the cookies [57], and they suggest that use of insect and certain plant oils in food products require additional research to improve their nutritional benefits. 

Volatile organic compounds (VOCs) constitute a major part of food aroma and they play a key role in food preference and acceptability. Our results reveal that the composition of the VOCs identified from the various plant oils corroborates those identified in previous studies [58,59,60,61]. For example, we detected 38 VOCs in olive oil which is within the previously reported numbers of (19–256) [58,59,60] and 29 VOCs in sesame oil compared to (17–221) [61]. However, these differences in VOCs levels may be associated with the sources of the oils and methods used to capture and analyze the volatiles. We found considerable variation in the composition of the aroma of the oils and their respective cookies. Perhaps for the plant oils, cultivar, geographical origin, freshness of oil, extraction techniques applied, ripening stage of the fruits, extraction conditions, and seed processing could account for this variation [58,59,60,61]. Previous studies on insect oils and fats have largely focused on proteins, fats, minerals, vitamins, amino acids, fatty acids, sterols, anti-nutritional factors, aroma of ready-to-eat whole insect products under different processing methods and partial substitute as ingredients in bakery products [4,6,8,23,62,63,64]. To the best of our knowledge, this is the first comprehensive study highlighting aroma, taste, color, smell, and complete inclusion of an insect oil as a key ingredient in a bakery product. These attributes are prerequisites for improved consumer acceptance.

Of the VOCs, we detected (*E*)-hex-2-enal, which is among the top five VOC markers used in establishing the freshness and quality of olive oil. We also detected related VOCs including C-6 alcohols, C-5 ketones, and esters, confirming the sensitivity and suitability of our volatile trapping and analysis methods. The presence of (*E*)-hex-2-enal in olive oil has been associated with sensory properties such as sweetness, fragrance, almond, green, and leafy taste [59]. Surprisingly, although (*E*)-hex-2-enal was detected in the volatiles of the cookies prepared with olive oil, it was one of the least preferred in terms of aroma and taste. This suggests that other components (parent oil and breakdown components due to baking) in the background aroma may contribute to the overall sensory attributes and consumer acceptance of the cookies.

The disappearance and appearance of compounds in food products are known to influence the sensory attributes such as aroma and taste and consumer acceptance of foods. These changes have been widely documented to be associated with the ingredients and conditions used in the preparation of the food items. In the present study, the quality of the dough, oil, and high temperatures [62,65] applied for the baking process may contribute to these changes. Notably, is the conspicuous loss of the two short chain fatty acids 3-methylbutanoic acid and 2-methylbutanoic acid in the cookies prepared from the insect oils. These two fatty acids are known to contribute a sweaty taste in foods such as French fries [66]. The appearance of new compounds like ketones responsible for the sweet and fruit flavor in food products during the baking process has also been previously demonstrated [67]. In the current study, the higher preference for cookies baked with plant than insect oils may be associated with the differential levels of the ketones 2,5-dimethyl-3-hexanone, 2-heptanone, and 2,5-octanedione detected in the aroma of these cookies. Likewise, detection of the aldehydes hexanal, heptanal, (2*E*,4*E*)-dodeca-2,4-dienal, and nonanal in the aroma of the different cookies may account for their overall acceptance by consumers in terms of their fat, meaty flavor, and almond-like aroma contents [65,68]. These results suggest that insect oils could be used for preparing other food products, which would require additional research. 

### 4.2. Sensory Evaluation and Consumer Acceptability

Our dietary evaluation showed that the cookies prepared from *S. gregaria* oil had significantly higher crude protein content than the other cookies. In sub-Saharan Africa the desert locust is an important source of nutrient-rich food. As such, our results agree with a previous finding, which showed that locusts are a rich source of protein (50.8–65.4% dry weight) and fat (34.93%) [69], *Locusta migratoria*, 50% to 65% [70,71], and generally for other orthopterans, ranging from 15% to 81% [72]. Orthopteran species are also known to contain satisfactory ratios of essential amino acids recommended for human consumption, making them a sustainable protein option for consumers looking for alternatives to traditional meat sources such as beef [73]. Therefore, the finalization of policies, guidelines and legislations developed for use of insect as food and feed should be accelerated and made operational to safeguard the consumer health and safety [74,75,76,77].

## 5. Conclusions

Our studies have demonstrated for the first time the comparative analysis of the composition of oils isolated from two commonly consumed grasshopper species *R. differens* and *S. gregaria* in sub-Saharan Africa. Cookies prepared with these two insect oils as substitutes for traditional butter appeared not to change the color, but rather their taste among panelists with no experience in eating insects (entomophagy). Although, insects and insect products are of high demand in many countries as a sustainable alternative to traditional animal-based food sources, we found that hedonic testing is paramount to decrease insect-based food products neophobia, as it entices people to ‘‘take the first step” with high likelihoods of future adaptation and acquaintance with entomophagy. Significant differences were observed for some sensory attributes and acceptability associations among bakery products with the oils of both insects. The high acceptability among consumers of cookies baked with the oil of *R. differens* indicates that future research should explore the use of refined or flavored insect oils to improve their overall acceptance in bakery products and other foods. Thus, the development and promotion of insects and insects-based food products for human consumption would generate interesting environmental and economic benefits for communities in Africa. 

## Figures and Tables

**Figure 1 foods-09-00800-f001:**
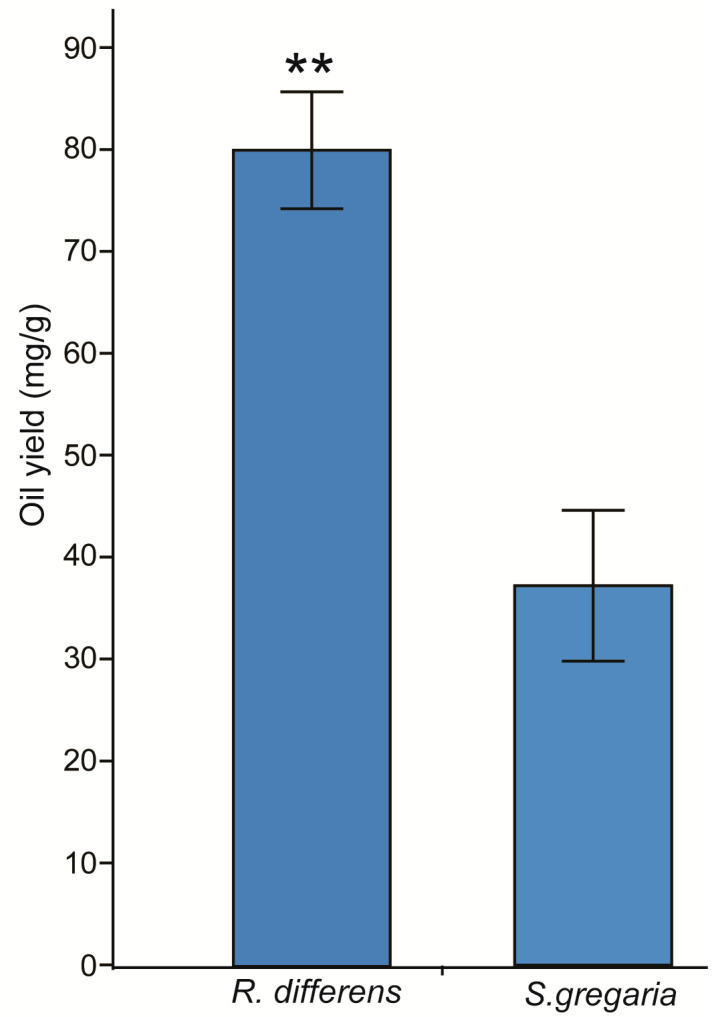
Bar chart showing the variation of oil yield between *R. differens* and *S. gregaria*. Error bars indicate the standard error of the mean. Asterisks indicate significant difference between the means: ** *p* < 0.01.

**Figure 2 foods-09-00800-f002:**
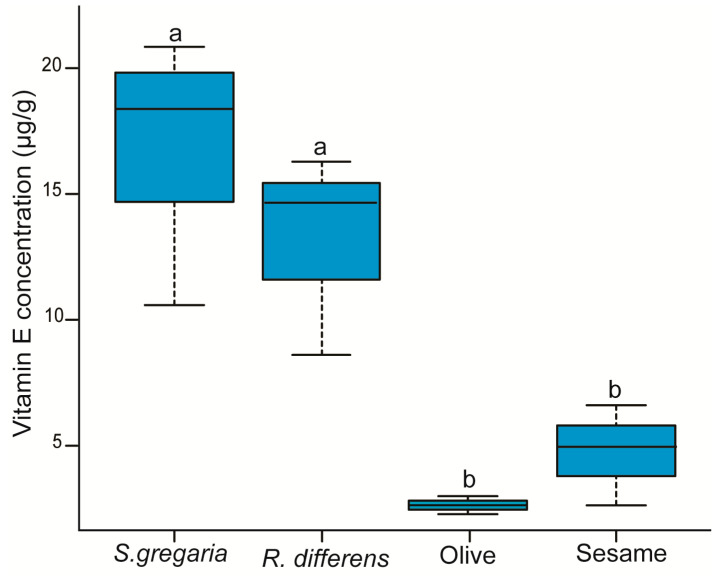
Boxplot showing vitamin E concentration in the different insect and plant oils. On each boxplot, the minimum and maximum values of all the data are represented by the ends of boxplot whiskers. Box plots with different letters are significantly different from each other (ANOVA followed by Student–Neuman–Keul’s (SNK’s) post-hoc test; *p* < 0.05, *n* = 3).

**Figure 3 foods-09-00800-f003:**
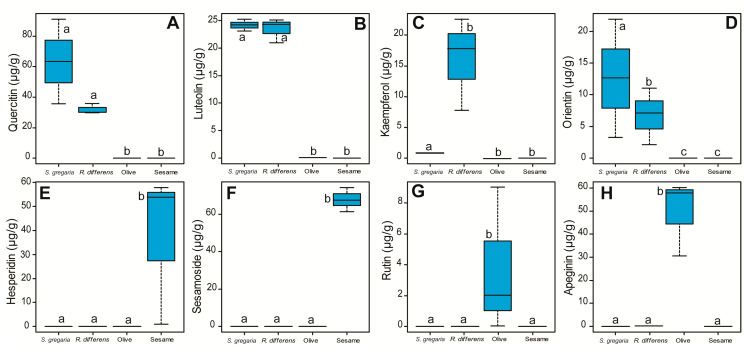
Flavonoid concentrations in the oils of different insect and plant species analyzed by Liquid Chromatography tandem Mass Spectrometry (LC-MS/MS). (**A**) quercetin, (**B**) luteolin, (**C**) kaempferol, (**D**) orientin, (**E**) hesperidin, (**F**) sesamoside, (**G**) rutin, and (**H**) apigenin. On each boxplot, the minimum and maximum values of all the data are represented by the ends of boxplot whiskers. Box plots with different letters are significantly different from each other (ANOVA followed by SNK’s post hoc test; *p* < 0.05, *n* = 3).

**Figure 4 foods-09-00800-f004:**
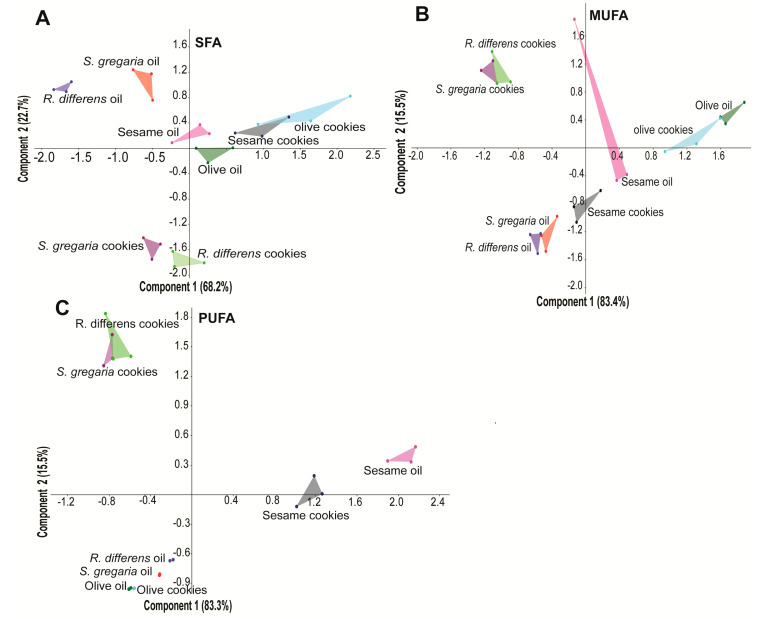
Principal component analysis (PCA) biplots showing the variation of (**A**) saturated fatty acid (SFA), (**B**) monounsaturated fatty acid (MUFA), and (**C**) polyunsaturated fatty acid (PUFA) among the different oil categories and cookies type.

**Figure 5 foods-09-00800-f005:**
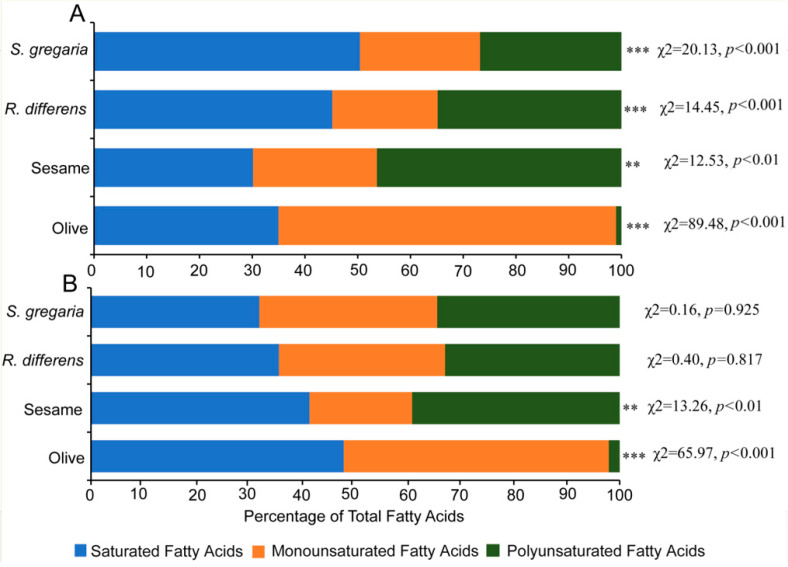
Fatty acid methyl esters (FAMEs) composition in different plants, insect oils and cookies prepared from the same oils. (**A**). FAMEs in plant and insect oils. (**B**). FAMEs in cookies prepared from the plant and insect oils. ** Denotes significantly different at 0.01. *** Denotes significantly different at 0.001.

**Figure 6 foods-09-00800-f006:**
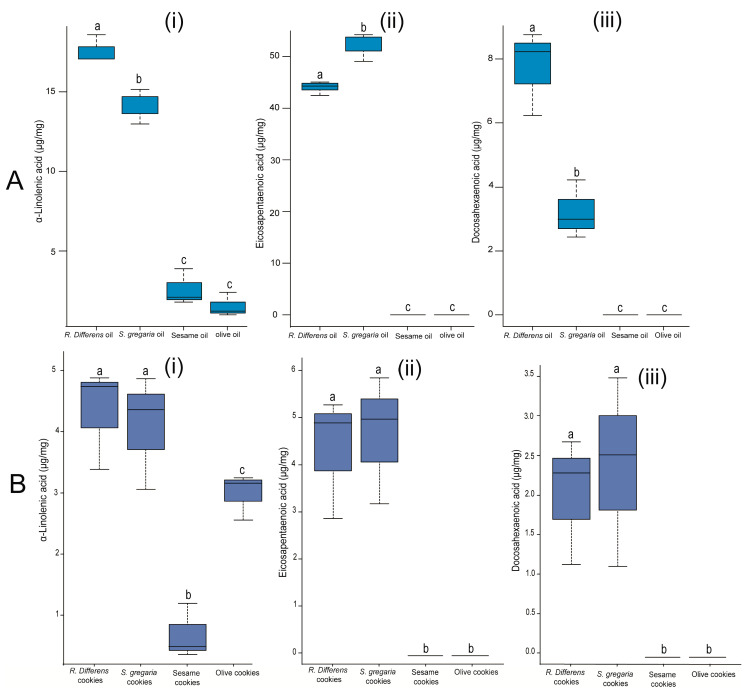
Boxplot illustrating the variation of (**i**) α-linolenic acid, (**ii**) eicosapentaenoic acid, and (**iii**) docosapentaenoic acid concentrations across the different oil (**A**) and cookie types (**B**). On each boxplot, the minimum and maximum values of all the data are represented by the ends of boxplot whiskers. Treatments with different lowercase letters are significantly different from each other.

**Figure 7 foods-09-00800-f007:**
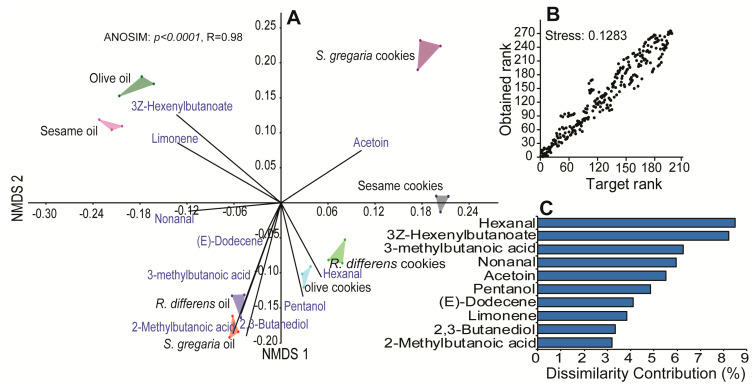
(**A**) Non-metric multidimensional scaling plot (NMDS) clustering the different oil categories based on the type of volatile they emit, analysis of similarities (ANOSIM) (**B**) Shepard plot showing the great ordination of the NMDS analysis (stress value <0.2). (**C**) Histogram depicting the contribution of the 10 most important volatiles to the differentiation of all the oil types.

**Figure 8 foods-09-00800-f008:**
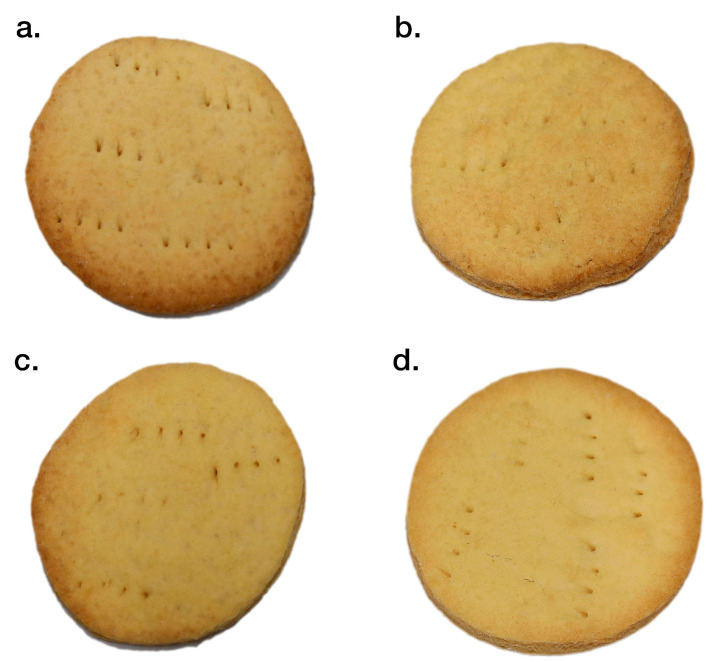
Wheat cookies samples prepared with (**a**) *R. differens*, (**b**) *S. gregaria*, (**c**) olive, and (**d**) sesame oils.

**Figure 9 foods-09-00800-f009:**
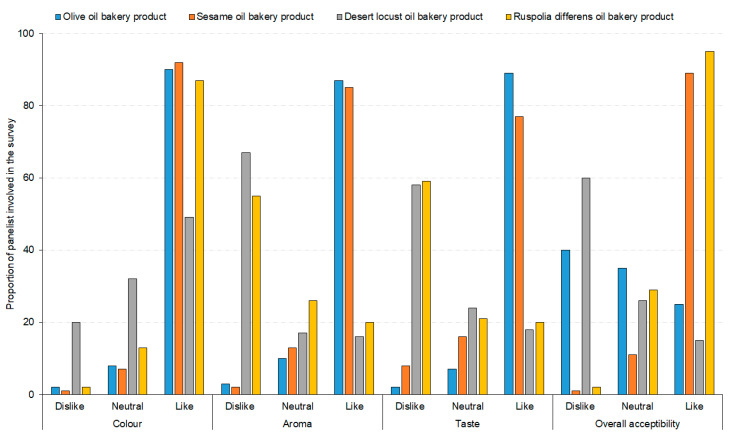
Proportion of panelists preference for color, aroma, taste, and overall acceptability of cookies fortified with insect-based oils.

**Figure 10 foods-09-00800-f010:**
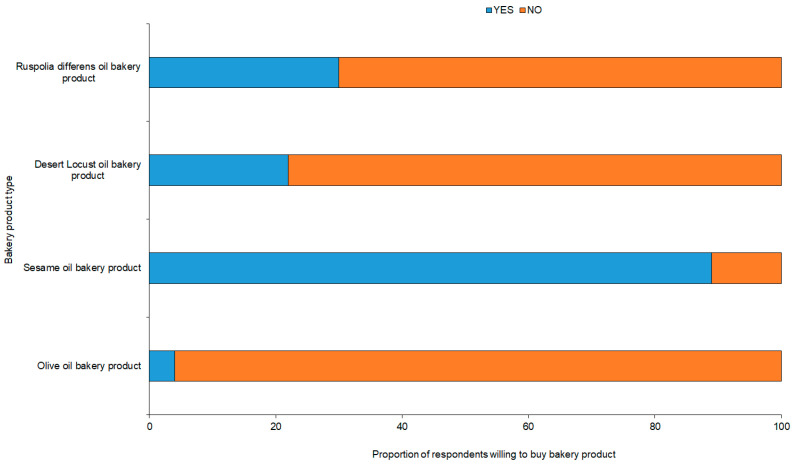
The percentage of panelist willing to buy cookies prepared with insect-based oils.

**Table 1 foods-09-00800-t001:** Fatty acid composition (µg/mg of oil and cookies ^a^) made from different plant and insect oils analyzed by Gas Chromatography coupled to Mass Spectrometry (GC-MS).

Peak No.	tR (min)	Compound Name	ω-n (Δn)	*S. gregaria* Oil	*R. differens* Oil	Sesame Oil	Olive Oil	*S. gregaria* Cookies	*R. differens* Cookies	Sesame Oil Cookies	Olive oil Cookies
1	13.67	Methyl octanoate	C8:0	1.5 ± 0.08	2.2 ± 0.39	1.7 ± 0.79		2.0 ± 0.16	2.1 ± 0.21		0.1 ± 0.02
2	16.44	Methyl decanoate	C10:0	0.2 ± 0.05	0.6 ± 0.17	2.0 ± 0.81		1.6 ±0.17	1.5 ± 0.19		0.2 ± 0.03
3	18.98	Methyl dodecanoate	C12:0	0.6 ± 0.12	0.9 ± 0.31	0.1 ± 0.03	0.1 ± 0.01	2.0 ± 0.18	1.8 ± 0.08	3.5 ± 0.24	2.3 ± 0.12
4	19.25	Methyl 2,6-dimethylundecanoate	iso-dimethyl-C11:0	0.2 ± 0.05	0.3 ± 0.11						
5	19.60	Methyl 11-methyldodecanoate	iso-methyl-C12:0	0.5 ± 0.13	0.4 ± 0.10						
6	19.69	Methyl 10-methyldodecanoate	iso-methyl-C12:0	0.3 ± 0.07	0.3 ± 0.06						
7	20.17	Methyl tridecanoate	C13:0	0.4 ± 0.07	1.0 ± 0.13			1.6 ± 0.23	1.5 ± 0.33		
8	20.46	Methyl 3-methyltridecanoate	iso-methyl-C13:0	0.1 ± 0.03	0.2 ± 0.07			0.01 ± 0.01			
9	20.75	Methyl 12-methyltridecanoate	iso-methyl-C13:0	0.2 ± 0.04	0.3 ± 0.08			0.02 ± 0.01			
10	21.32	Methyl tetradecanoate	C14:0	8.6 ± 1.50	19.2 ± 1.56	0.9 ± 0.09	1.0 ± 0.05	2.3 ± 0.25	1.89 ± 0.32	14.9 ± 1.14	12.5 ± 0.66
11	21.51	Methyl 2-methyltetradecanoate	iso-methyl-C14:0	0.3 ± 0.07	0.7 ± 0.21			0.9 ± 0.02			
12	21.84	Methyl 13-methyltetradecanoate	iso-methyl-C14:0	0.7 ± 0.12	1.2 ± 0.16			0.2 ± 0.04			
13	21.93	Methyl 9-methyltetradecanoate	iso-methyl-C14:0	1.3 ± 0.33	2.2 ± 0.08			0.4 ± 0.10			
14	22.37	Methyl pentadecanoate	C15:0	1.7 ± 0.25	3.2 ± 0.37	0.1 ± 0.01	0.5 ± 0.03	1.7 ± 0.25	1.78 ± 0.14	1.1 ± 0.07	1.1 ± 0.09
15	22.66	Methyl 3-methylpentadecanoate	iso-methyl-C15:0	0.1 ± 0.03	0.3 ± 0.02						
16	23.66	Methyl hexadecanoate	C16:0	171.7 ± 26.93	73.6 ± 4.70	171.7 ± 9.44	197.2 ± 9.62			295.2 ± 22.56	250.4 ± 13.12
17	23.60	Methyl 10-hexadecanoate	iso-methyl-C16:0	71.0 ± 16.58	89.2 ± 1.80				174.8 ± 8.93		
18	24.05	Methyl heptadecanoate	C17:0			4.9 ± 0.59	10.5 ± 1.33	139.7 ± 4.39	90.9 ± 4.64	3.1 ± 0.24	2.6 ± 0.14
19	24.14	Methyl 14-methylhexadecanoate	iso-methyl-C16:0	0.7 ± 0.07	0.9 ± 0.19			72.1 ± 2.21	5.86 ± 0.36		
20	25.55	Methyl octadecanoate	C18:0	51.8 ± 3.27	58.5 ± 1.14	92.1 ± 4.15	61.9 ± 1.15	3.8 ± 1.11	45.4 ± 2.32	88.5 ± 6.77	66.9 ± 3.51
21	25.62	Methyl 6-methyloctadecanoate	iso-methyl-C18:0	0.4 ± 0.08	0.9 ± 0.17			41.7 ± 1.43			
22	26.21	Methyl nonadecanoate	C19:0	0.6 ± 0.08	1.6 ± 0.17	1.1 ± 0.07	1.8 ± 0.08	3.2 ± 0.31	3.4 ± 0.37	0.8 ± 0.07	0.6 ± 0.05
23	27.05	Methyl eicosanoate	C20:0			30.1 ± 1.08	23.2 ± 1.71				10.0 ± 0.53
24	27.17	Methyl 2,6-dimethylnonadecanoate	iso-dimethyl-C19:0	0.2 ± 0.06	1.1 ± 0.45						
25	27.86	Methyl 2,4-dimethylheneicosanoate	iso-dimethyl-C22:0	0.9 ± 0.69	0.1 ± 0.07						
26	27.87	Methyl heneicosanoate	C21:0	0.1 ± 0.03	0.5 ± 0.19	1.1 ± 0.10	2.6 ± 0.47	2.4 ± 0.22	2.2 ± 0.21		0.6 ± 0.05
27	28.01	Methyl 3-methylheneicosanoate	iso-methyl-C22:0	0.07 ± 0.02	0.4 ± 0.11						
28	28.66	Methyl docosanoate	C22:0			8.9 ± 0.43	8.7 ± 0.73	5.5 ± 0.29	4.6 ± 0.53	3.4 ± 0.26	2.3 ± 0.12
29	29.42	Methyl tricosanoate	C23:0			2.1 ± 0.31	2.8 ± 0.23	2.2 ± 0.38	2.0 ± 0.24	2.0 ± 0.16	1.4 ± 0.19
30	30.06	Methyl 16-methyltricosanoate	iso-methyl-C23:0								
31	30.21	Methyl tetracosanoate	C24:0			6.5 ± 0.42	5.0 ± 0.62	1.6 ± 0.24	0.9 ± 0.20	4.3 ± 0.30	0.4 ± 0.03
32	31.10	Methyl pentacosanoate	C25:0			2.1 ± 0.24					1.9 ± 0.21
33	32.17	Methyl hexacosanoate	C26:0			2.2 ± 0.49	1.1 ± 0.36	1.3 ± 0.12	1.4 ± 0.21	1.9 ± 0.16	0.8 ± 0.06
34	33.41	Methyl 20-methylhexacosanoate	iso-methyl-C26:0				0.2 ± 0.01				
35	34.93	Methyl octacosanoate	C28:0			0.9 ± 0.19	0.2 ± 0.05				1.2 ± 0.25
		Ʃ SFA		297.6 ± 26.18	259.9 ± 12.79	328.4 ±19.25	316.8 ± 16.43	285.5 ± 11.09	341.7 ± 19.29	355.5 ± 19.18	418.58 ± 31.95
36	14.27	Methyl 2-hexenoate	C6:1 (n-4)	0.1 ± 0.02	0.07 ± 0.02						
37	20.88	Methyl (11*E*)-tetradecenoate	C14:1 (n-3)	0.3 ± 0.04	0.3 ± 0.08			0.03 ± 0.01			
38	20.94	Methyl (11*Z*)-tetradecenoate	C14:1 (n-3)	0.5 ± 0.12	0.3 ± 0.19			0.1 ± 0.01			
39	21.01	Methyl (9*E*)-dodecenoate	C14:1 (n-3)			0.02 ± 0.01					0.02 ± 0.01
40	21.02	Methyl (9*Z*)-tetradecenoate	C14:1 (n-3)			0.1 ± 0.02	0.1 ± 0.01		1.3 ± 0.27		0.5 ± 0.09
41	22.10	Methyl (5*Z*)-decenoate	C10:1 (n-5)			0.3 ± 0.04	0.85 ± 0.08				0.9 ± 0.13
42	23.24	Methyl (9*Z*)-hexadecenoate	C16:1 (n-7)	3.1 ± 0.27	5.9 ± 0.45	5.9 ± 0.26	27.7 ± 1.34	2.0 ± 0.26		5.3 ± 0.37	4.6 ± 0.24
43	24.01	Methyl 8-heptadecenoate	C17:1 (n-9)	5.04 ± 0.22	8.8 ± 0.64						
44	24.19	Methyl (10*Z*)-heptadecenoate	C17:1 (n-7)	1.4 ± 0.22	2.7 ± 0.37	2.4 ± 0.08	10.0 ± 0.94	3.5 ± 0.42	3.8 ± 0.29	1.6 ± 0.13	2.1 ± 0.28
45	24.91	Methyl (6*Z*)-octadecenoate	C18:1 (n-12)	0.6 ± 0.08	1.3 ± 0.14						
46	25.24	Methyl (9*Z*)-Octadecenoate	C18:1(n-9)	53.8 ± 2.48	41.1 ± 1.81	230.9 ± 13.93	514.4 ± 15.08	1.5 ± 0.14	2.7 ± 0.31	425.8 ± 32.54	151.2 ± 17.30
47	25.25	Methyl (6*E*)-octadecenoate	C18:1 (n-12)	6.1 ± 0.13	10.7 ± 0.74						
48	25.31	Methyl 16-octadecenoate	C18:1 (n-2)	61.9 ± 0.80	40.2 ± 1.31						
49	25.38	Methyl (9*E*)-octadecenoate	C18:1(n-9)				0.7 ± 0.01	206.8 ± 6.65	205.5 ± 10.49		
50	25.41	Methyl (3*Z*)-octadecenoate	C18:1 (n-15)					70.2 ± 3.40	69.6 ± 3.56		
51	26.01	Methyl (10*Z*)-nonadecenoate	C19:1 (n-9)			6.6 ± 3.15	6.0 ± 0.46	10.7 ± 0.37	11.1 ± 0.62		1.3 ± 0.07
52	26.72	Methyl (11*E*)-eicosenoate	C20:1(n-9)	1.1 ± 0.25	2.4 ± 0.35						
53	26.86	Methyl (11*Z*)-eicosenoate	C20:1(n-9)				19.3 ± 1.61	3.2 ± 0.32	3.9 ± 0.21	6.0 ± 0.46	
54	26.87	Methyl (13*Z*)-eicosenoate	C20:1(n-7)	0.3 ± 0.06	0.6 ± 0.11	8.7 ± 2.09					4.9 ± 0.26
55	28.47	Methyl (13*Z*)-docosenoate	C22:1(n-9)			0.7 ± 0.06	0.6 ± 0.01				0.6 ± 0.05
56	28.48	Methyl (11*Z*)-docosenoate	C22:1(n-11)			0.3 ± 0.05					0.4 ± 0.05
57	30.03	Methyl (15*Z*)-tetracosenoate	C24:1(n-9)					3.5 ± 0.66	3.7 ± 0.19		
		Ʃ MUFA		134.3 ± 4.69	114.5 ± 6.18	255.9 ± 19.67	579.3 ± 19.53	301.3 ± 11.20	301.5 ± 15.95	166.5 ± 18.48	438.8 ± 33.50
58	24.78	Methyl (9*Z*,11*E*)-octadecadienoate	C18:2(n-7)					2.9 ± 0.23	2.1 ± 0.38		
59	25.21	Methyl (9*Z*,12*Z*)-octadecadienoate	C18:2(n-6)	60.2 ± 0.19	83.7 ± 1.54	496.9 ± 12.67	4.0 ± 0.91	2.9 ± 0.23	2.1 ± 0.38	330.9 ± 11.54	11.5 ± 0.35
60	25.33	Methy (9*Z*,11*Z*)-octadecadienoate	C18:2(n-7)	0.6 ± 0.19	1.3 ± 0.14						
61	25.47	Methyl (5,12)-octadecadienoate	C18:2(n-7)					126.7 ± 3.34	130.9 ± 6.68		
62	25.60	Methyl (10*E*,12*Z*)-octadecadienoate	C18:2(n-6)	7.7 ± 0.19	15.4 ± 0.28		2.0 ± 0.12	144.6 ± 3.81	149.5 ± 7.63		
63	25.71	Methyl (3R,6*E*,10*E*)-3,7,11,15-tetramethylhexadeca-6,10,14-trienoate	iso-tetramethyl C20:3(n-1)	2.3 ± 0.19	4.4 ± 0.18						
64	25.71	Methyl (9*Z*,15*Z*)-octadecadienoate	C18:2(n-3)	0.9 ± 0.19	1.9 ± 0.08						
65	25.74	Methyl (7,10)-octadecadienoate	C18:2(n-8)	0.1 ± 0.08	0.2 ± 0.09	1.3 ± 0.11					1.1 ± 0.9
66	25.90	Methyl (9*E*,12*E*)-octadecadienoate	C18:2(n-6)	2.2 ± 0.19	5.6 ± 0.25	2.7 ± 0.37		7.8 ± 0.37	7.9 ± 0.46	2.3 ± 0.16	2.1 ± 0.29
67	26.24	Methyl (9*E*,11*E*)-octadecadienoate	C18:2(n-7)	4.2 ± 0.19							
68	26.34	Methyl (9*Z*,12*Z*,15*Z*)-octadecatrienoate (ALA)	C18:3(n-3)	14.1 ± 0.51	17.6 ± 0.41	2.6 ± 0.54	1.7 ± 0.28	4.1 ± 0.44	4.3 ± 0.39	0.7 ± 0.21	1.3 ± 0.09
69	26.40	Methyl (5*Z*,8*Z*,11*Z*,14*Z*-eicosatetraenoate	C20:4(n-6)	4.1 ± 0.19	7.4 ± 0.58						
70	26.45	Methyl (9*Z*,11*E*,13*E*)-octadecatrienoate	C18:3(n-3)	3.0 ± 0.19	5.3 ± 0.55	1.3 ± 0.10	1.7 ± 0.28	16.0 ± 0.84	16.2 ± 1.00	1.3 ± 0.09	2.5 ± 0.27
71	26.54	Methyl (5*Z*,8*Z*,11*Z*,14*Z*,17*Z*)-eicosapentaenoate (EPA)	C20:5(n-3)	52.1 ± 1.29	44.0 ± 0.65			4.7 ± 0.64	4.3 ± 0.61		
72	26.68	Methyl (11*E*,14*E*)-eicosadienoate	C20:2(n-6)	3.5 ± 0.19	5.9 ± 0.78						
73	28.12	Methyl (4*Z*,7*Z*,10*Z*,13*Z*,16*Z*,19*Z*)-docosahexaenoate (DHA)	C22:6(n-3)	3.2 ± 0.43	7.7 ± 0.63			2.4 ± 0.56	2.0 ± 0.38		
		Ʃ PUFA		158.2 ± 4.22	200.5 ± 6.17	505.0 ± 13.78	9.3 ± 1.67	324.8 ± 12.59	333.8 ± 19.33	337.3 ± 12.39	18.1 ± 0.78
		Ʃ n-6 PUFA		73.6 ± 0.80	118.1 ± 3.43	49.9 ± 6.52	6.0 ± 1.06	294.8 ± 9.88	304.9 ± 16.58	33.3 ± 2.52	13.8 ± 0.51
		Ʃ n-3 PUFA		74.2 ± 2.8	77.3 ± 2.59	4.0 ± 1.12	3.2 ± 0.64	27.2 ± 2.49	26.8 ± 2.38	3.2 ± 0.95	4.3 ± 0.27
		Ʃ n-6/n-3		1.4	1.5	12.5	1.9	10.8	11.35	10.4	3.2
		Ʃ ALA+ EPA + DHA		69.5 ± 2.23	69.3 ± 1.69	2.6 ± 0.53	1.5 ± 0.36	11.1 ± 2.85	10.7 ± 1.38	0.7 ± 0.21	3.0 ± 0.17

(tR Retention time, ^a^ Mean ± SE (standard error) of triplicate determinations). SFA saturated fatty acids, MUFA monounsaturated fatty acids, PUFA polyunsaturated fatty acids, ALA α-linolenic acid, EPA eicosapentaenoic acid, DHA docosapentaenoic acid.

**Table 2 foods-09-00800-t002:** Identified flavonoids in analyzed oils.

Peak No.	tR (Min)	Flavonoid Name	Molecular Formula	[M + H] ^+^	Key Fragment Ions	*S. gregaria* Oil	*R differens* Oil	*Sesame* Oil	*Olive* Oil
1 ^a^	6.20	Rutin	C_27_H_30_O_16_	611.2	633.4, 465.3, 303.3	−	−	+	+
2 ^a^	6.39	Apigenin	C_15_H_10_O_5_	271.2	293.4, 153.5, 121.4, 145.6	−	−	−	+
3 ^a^	7.98	Quercetin	C_15_H_10_O_7_	303.4	325.2, 285.1, 229.3, 153.2, 137.5	+	+	−	−
4 ^a^	8.79	Luteolin	C_15_H_10_O_6_	287.3	309.3, 269.1, 153.2, 135.2	+	+	−	−
5 ^a^	9.60	Kaempferol	C_15_H_10_O_6_	287.5	309.6, 153.4, 137.3, 121.6	+	+	−	−
6 ^b^	12.30	Orientin	C_21_H_20_O_11_	449.3	471.2, 299.1, 329.2	+	+	−	−
7 ^b^	12.62	Hesperidin	C_28_H_34_O_15_	611.5	633.2, 593.3, 575.2, 303.4	−	−	+	−
8 ^b^	12.73	Sesamoside	C_17_H_24_O_12_	421.4	863.4, 459.2, 443.2	−	−	+	−

tR Retention time; ^a^ Flavonoids identity confirmed with authentic standard; ^b^ Flavonoids tentatively identified; + = present and − = not detected.

**Table 3 foods-09-00800-t003:** Volatile organic compounds (ng/g/h ^a^) of the selected oils from insects, plants, and cookies analyzed by GC-MS.

tR (min)	Compound	Compound Class	*S. gregaria* Oil	*R. differens* Oil	Sesame Oil	Olive Oil	*S. gregaria* Cookies	*R. differens* Cookies	Sesame Oil Cookies	Olive Oil Cookies
3.43	1-Penten-3-ol	Alcohol	8.9 ± 3.26	17.2 ± 1.33		5.4 ± 0.46			3.9 ± 0.55	19.0 ± 0.89
3.68	Pentanal	Aldehyde	20.8 ± 0.93			11.9 ± 1.05				18.6 ± 0.81
3.92	Acetoin	Ketone	46.6 ± 2.95				73.5 ± 1.60			
4.61	3-Methyl-1-butanol	Alcohol		4.5 ± 0.42						
4.71	3-Penten-2-one	Ketone								
4.94	2-Methylpentanal	Aldehyde				3.7 ± 1.36				
5.46	2,3-Dimethylhexane	hydrocarbon			26.9 ± 3.33	14.1 ± 1.38				
5.58	Pentanol	Alcohol		27.2 ± 1.11		4.0 ± 1.36		30.2 ± 1.20		
5.96	2,3-Butanediol	Alcohol	70.0 ± 1.98	10.8 ± 0.44						
6.49	Hexanal	Aldehyde	74.3 ± 2.41	18.4 ± 0.74	43.2 ± 4.83	27.9 ± 2.52	54.2 ± 2.72	79.8 ± 3.33	37.5 ± 4.16	163.3 ± 9.91
6.67	2-Hexanol	Alcohol				4.1 ± 0.37				
7.15	Methylpyrazine	nitrogen comp.		3.1 ± 0.24			6.8 ± 0.24	5.7 ± 0.15		
7.40	(*Z*)-1-Methoxyhex-3-ene	Ester				12.6 ± 1.03				
7.44	Furfural	Aldehyde			4.7 ± 0.15	5.9 ± 1.39	12.2 ± 0.73	8.3 ± 0.26	5.2 ± 1.25	15.1 ± 1.00
7.55	2,5-Dimethyl-3-hexanone	Ketone								5.7 ± 0.51
7.62	2,4-Dimethyl-1-heptene	hydrocarbon				17.0 ±0.64			4.7 ± 0.37	12.9 ± 0.83
7.93	Ethyl 2-methylbutanoate	Ester				4.2 ± 1.22				4.5 ± 0.36
7.97	(*E*)-2-Hexenal	Aldehyde				18.2 ± 0.52				4.5 ± 0.13
8.04	(*E*)-3-Hexenol	Alcohol				57.2 ± 2.88				
8.04	3-Furanmethanol	furanoid								12.5 ± 0.63
8.10	Ethylbenzene	benzenoid			12.4 ± 1.53	12.2 ± 1.34	11.8 ± 0.50			
8.27	(*Z*)-3-Hexen-1-ol	Alcohol				7.5 ± 0.37				
8.32	*p*-Xylene	benzenoid	5.2 ± 2.23		21.3 ± 0.47	12.5 ± 1.18			4.7 ± 0.15	5.7 ± 0.18
8.32	3-Methylbutanoic acid	Carboxylic acid	137.6 ± 7.08	19.4 ± 2.07						
8.38	Hexanol	Alcohol				38.0 ± 1.54				6.3 ± 0.38
8.62	2-Methylbutanoic acid	Carboxylic acid	50.8 ± 1.19	29.2 ± 1.69						
8.83	Styrene	benzenoid	8.7 ± 3.11	9.7 ± 1.53	17.8 ± 0.39	15.9 ± 1.82	3.1 ± 0.11	5.4 ± 0.20	3.1 ± 0.05	3.4 ± 0.03
8.90	2-Heptanone	Ketone		11.4 ± 0.63				12.6 ± 0.64	4.8 ± 0.13	15.3 ± 0.80
9.09	(*Z*)-4-Heptenal	Aldehyde								5.3 ± 0.44
9.14	Heptanal	Aldehyde		7.8 ± 0.43	9.8 ± 0.40			12.6 ± 0.64	8.3 ± 0.15	21.9 ± 2.66
9.24	Methional	Aldehyde								4.2 ± 0.45
9.33	2,5-Dimethylpyrazine	nitrogen comp.	6.2 ± 0.59	10.7 ± 1.59				7.9 ± 0.35	2.9 ± 0.04	13.3 ± 1.30
9.66	(1*Z*,5*E*)-Cycloocta-1,5-diene	hydrocarbon								3.6 ± 0.47
9.82	α-Pinene	monoterpene		3.5 ± 0.25	12.5 ± 1.63	6.8 ± 0.78		4.2 ± 0.12	4.2 ± 0.26	8.0 ± 1.36
10.08	Pyrrolidine	nitrogen comp.								12.8 ± 1.38
10.14	1-Methyl-2-propylcyclohexane	hydrocarbon	6.4 ± 1.55		8.1 ± 0.79	3.5 ± 0.47		7.6 ± 0.33		9.9 ± 1.03
10.33	(*E*)-2-heptenal	Aldehyde	27.2 ± 0.80	10.0 ± 0.55	9.6 ± 0.86					15.3 ± 1.55
10.39	Benzaldehyde	Aldehyde	12.6 ± 0.57	10.8 ± 1.59	34.8 ± 2.00			25.6 ± 1.44	8.2 ± 0.70	33.7 ± 1.93
10.73	δ-3-Carene	monoterpene				4.9 ± 0.45				11.0 ± 1.34
10.73	Myrcene	monoterpene							3.0 ± 0.09	7.1 ± 0.12
10.80	1-Octen-3-ol	Alcohol	20.5 ± 1.68	5.9 ± 0.34						21.8 ± 1.21
10.84	Phenol	benzenoid		5.2 ± 0.31	13.5 ± 0.69	5.0 ± 0.88				
10.91	2,5-Octanedione	Ketone								10.9 ± 1.34
11.00	Methoxymethylbenzene	benzenoid				8.1 ± 0.69				
11.04	2-Pentyl furan	furanoid					15.5 ± 0.82	12.9 ± 0.66	7.2 ± 0.58	16.1 ± 0.35
11.07	1,2,4-Trimethylbenzene	furanoid	5.1 ± 1.65	6.9 ± 0.63	26.4 ± 1.49	10.1 ± 1.84	6.1 ± 0.24			
11.24	2,3,5-Trimethylpyrazine	nitrogen comp.								27.0 ± 0.65
11.28	Octanal	Aldehyde				8.1 ± 1.14			6.7 ± 0.37	21.0 ± 0.98
11.34	(*Z*)-3-Hexenylbutanoate	Ester				183.7 ± 15.10				
11.37	3,3,5-Trimethylheptane	hydrocarbon		9.7 ± 0.95	31.1 ± 1.78					
11.45	2*E*,4*E*-Heptadienal	Aldehyde	6.8 ± 0.53					8.3 ± 0.26		
11.64	1-Undecyne	hydrocarbon				10.3 ± 1.09	15.4 ± 0.55	9.9 ± 0.33	5.8 ± 0.22	51.2 ± 1.15
11.75	Limonene	monoterpene		15.4 ± 1.84	85.9 ± 3.59		14.6 ± 0.76	6.5 ± 0.18	5.4 ± 0.43	11.9 ± 0.41
11.91	(*Z*)-β-Ocimene	monoterpene			15.3 ± 0.56					
12.02	1-Tetradecyl acetate	Ester			15.5 ± 0.27					
12.04	Benzene acetaldehyde	Aldehyde	6.9 ± 0.58	11.9 ± 1.36				7.5 ± 0.23		17.7 ± 0.66
12.10	(*E*)-β-Ocimene	monoterpene			25.1 ± 1.98	8.6 ± 1.70				
12.56	(Z)-9-Methyl-5-undecene	hydrocarbon			20.9 ± 1.15	15.2 ± 0.95			5.4 ± 0.20	12.0 ± 0.42
12.77	(*E*)-Dodecene	hydrocarbon			15.2 ± 0.32	12.3 ± 0.67				94.5 ± 5.67
13.07	Nonanal	Aldehyde	20.1 ± 0.67	30.7 ± 3.88	111.3 ± 4.67			31.7 ± 1.24		
13.80	Camphor	monoterpene			14.0 ± 0.73	4.9 ± 0.94				
14.41	Naphthalene	benzenoid			11.2 ± 0.22	4.4 ± 0.71				
14.76	Decanal	Aldehyde								9.0 ± 0.68
15.43	Precocene I	chromene	3.5 ± 2.11	4.8 ± 0.45						
16.20	(2*E*,4*E*)-Dodeca-2,4-dienal	Aldehyde					3.2 ± 0.06	4.1 ± 0.08	2.7 ± 0.03	8.2 ± 0.26
17.11	Butyl butanoate	Ester		5.4 ± 0.32	28.4 ± 1.12	12.2 ± 1.73				
17.20	α-Copaene	sesquiterpene			9.5 ± 0.31					
17.27	Pentadecanol	Alcohol	4.3 ± 1.52	4.7 ± 0.29						4.7 ± 0.11
17.65	Longifolene	sesquiterpene			11.7 ± 0.40	5.9 ± 0.64				
17.73	(*Z*)-α-Bisabolene	sesquiterpene			11.1 ± 0.55	6.0 ± 0.54				
18.82	2,4-bis(1,1-dimethylethyl)-phenol	benzenoid	8.9 ± 1.64	10.7 ± 0.44	27.0 ± 1.03	15.9 ± 1.36		7.7 ± 0.23		3.8 ± 0.07
18.85	4-tert-butylphenyl acetate	Ester					5.6 ± 0.21		3.9 ± 0.07	
18.96	*o*-Hydroxybiphenyl	benzenoid		4.5 ± 0.26	24.2 ± 0.94	9.6 ± 1.52				

(tR Retention time, ^a^ Mean ± SE (standard error) of triplicate determinations).

**Table 4 foods-09-00800-t004:** Proximate analysis (% dry matter) * of cookies prepared using different oil sources.

Treatment	Proximate Analysis				
	Moisture	Ash	Fiber	Crude Protein	Carbohydrate
Desert locust oil (*S. gregaria*)	5.48 ± 0.84 ^a^	2.31 ± 0.13 ^a^	0.13 ± 0.001 ^b^	8.82 ± 0.22 ^a^	51.36 ± 0.79 ^c^
Olive oil	3.89 ± 0.29 ^b^	1.78 ± 0.18 ^b^	0.15 ± 0.003 ^a^	7.49 ± 0.37 ^b^	71.47 ± 0.61 ^b^
African bush-cricket oil (*R. differens*)	3.54 ± 0.07 ^b^	1.53 ± 0.01 ^c^	0.11 ± 0.002 ^c^	7.75 ± 0.28 ^b^	71.39 ± 0.39 ^b^
Sesame oil	3.24 ± 0.09 ^b^	1.50 ± 0.01 ^c^	0.05 ± 0.004 ^d^	7.48 ± 0.16 ^b^	73.86 ± 0.68 ^a^

* proximate analysis of cookies is expressed as it is, values with different lowercase letters are significantly different from each other.

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
