# Peer review of "Chemistry and Sensory Characterization of a Bakery Product Prepared with Oils from African Edible Insects"

_foods, 2020, doi:10.3390/foods9060800_

Round 1

Reviewer 1 Report

General:

The study by Cheseto et al. addresses the preparation of oils from two edible insects and characterises their chemical and nutritional components. The work is interesting for a comparison to plant oils and as a contribution to increase the scope of food resources. In addition, the work includes a costumer’s acceptance study for a bakery product made from the insect oils and compared to products based on plant oils.

The manuscript contains detailed chemical information, and is precise in the presentation on methods and the graphic and tabular representation of the findings.

The work seems suitable for publication in “Foods” while I also highlight some aspects that need attention for revision.

Major:

Title: change to “…sensory characterisation of a bakery product...” since you only run tests with one type of product, the cookie.

In some respects, the introduction/ discussion will benefit from some more specific background information. For example, in the introduction, information is given for different insects on fatty acids or fat content – but what about the species studied here, locusts and katydids, Schistocerca and Ruspolia? Have they not been investigated previously for body fat/ oil content and its components as well as its use as food resource? If available, add some references.

Also, the introduction and discussion seem to be scarce with references to the use of insect food from the area discussed here, sub-Saharan Africa. You could present the situation of insects, especially their oils, as food source in some more details, probably backed up by references from the recent book volume on “African Edible Insects As Alternative Source of Food, Oil, Protein and Bioactive Components“, or similar resources.

In the presentation, there are generally cross-relations between different components or parameters of cookies discussed, while occasionally some broader comparison would also be helpful.

Fig. 10 shows that there are the strongest dislikes associated with products including insect oils – for all 5 parameters gaining a dislike ratio >50% of panellists come from insect oil products. For a costumer’s acceptance study, this should be spelled out at least in the discussion and could be considered even for the abstract, as it may hint at a fundamental difference in the acceptabilty of plant over insect oils. The one positive exception is the overall acceptability for the bakery product with Ruspolia oil. However, there may also be different types of food which could be better suited for insect oils? This is only vaguely hinted at in the conclusion (l. 601), but maybe you could elaborate on other uses for the insect oils.

For another example, you also carried out chemical analysis of olive and sesame oils de novo, and compare the composition for FAs in the discussion. The overall sum of FAs present in your insect and plant samples is stated (p. 279). But what about the similarities – just include a short mention if there are major differences, substances unique to one type of oil etc. What about the earlier studies of plant oils you cite – did they miss something, or did you not find a component in your sample reported previously? Would this possibly relate to the method of extraction?

The discussion should be structured better by breaking it down into sub-sections, e. g. 1. on the differences in oil chemistry between insects and plants, 2. Details on contents and their nutritional and health effects, and 3. the analysis of consumer acceptance results.

Figures

Figure 7 is far too small – is there something missing in the file provided for review?

There seems to be a mistake by exchanging the figures 8 and 10 – please revert! I’d also suggest to move figure 9 up in place of figure 8, to give an impression of the cookies first which are then discussed in the following.

Tables

Abbreviations are given at the end of the table. Since some tables are very long and extend over several pages, please move the abbreviations up so they are next to the heading and easily identify the meaning within the table on a first look.

Minor:

l. 10 insect-derived ingredients

l. 12f knowledge… remains to be established.

l. 20 What do you mean by proximate analysis? Are there alternatives?

l. 21, 26 Consumer’s acceptance (it’s the acceptability of the cookie under scrutiny, not of the consumers)

l. 49 isn’t the emission of greenhouse gases and water and land requirement contributing to the ecological footprint?

l. 53 – 57 Please add references for the numbers cited here.

l. 100 delete comma

l. 101 delete “and used for the study.”

l. 212 It would be better if you keep with the parameters and their order given above (l. 200) to follow the analysis better, for example, the texture does not appear again later.

l. 435 …had a significantly higher crude protein content…

l. 447 panelist’s acceptance

l. 470 Obviously the oil yield from insects depends on several extrinsic (diet, temperature) or intrinsic (development, age) factors. Another aspect could be differences in species for the fat/oil deposition in the body. As R. differens has a high body content of fat (J. N. Kinyuru et al. 2018, The Role of Edible Insects in Diets and Nutrition in East Africa), are there similar numbers for comparison available from S. gregaria?

l. 508 …in living organisms including, in insects, it regulated… - something seems to be missing here in the argument – are two sentences fused?

l. 548 “cultivar ,” – delete free space

The abbreviations used for FAs are given in the methods section (l. 220) and later in the results (l. 279-281) but FAMEs are only included on the methods (l. 104). After an extensive method description, it would be helpful to include also the abbreviation for FAMEs again (l. 293, 320).

Author Response

Response to Reviewer 1 Comments

Point 1. General: The study by Cheseto et al. addresses the preparation of oils from two edible insects and characterizes their chemical and nutritional components. The work is interesting for a comparison to plant oils and as a contribution to increase the scope of food resources. In addition, the work includes a costumer’s acceptance study for a bakery product made from the insect oils and compared to products based on plant oils.

The manuscript contains detailed chemical information, and is precise in the presentation on methods and the graphic and tabular representation of the findings.

The work seems suitable for publication in “Foods” while I also highlight some aspects that need attention for revision.

Response 1: Thank you for the positive comments on our manuscript

Point 2. Title: change to “…sensory characterisation of a bakery product...” since you only run tests with one type of product, the cookie.

Response 2: Addressed (see text). The title has been changed to read ‘Chemistry and sensory characterization of a bakery product prepared with oils from African edible insects

Point 3. In some respects, the introduction/ discussion will benefit from some more specific background information. For example, in the introduction, information is given for different insects on fatty acids or fat content – but what about the species studied here, locusts and katydids, Schistocerca and Ruspolia? Have they not been investigated previously for body fat/ oil content and its components as well as its use as food resource? If available, add some references

Response 3: Addressed (see text). Information noted with thanks. The fat contents/oils of the two-insects, the desert locust Schistocerca gregaria and African bush-cricket Ruspolia differens have been added in lines 64-68 and two references added (Reference, [19,20]).

Point 4. Also, the introduction and discussion seem to be scarce with references to the use of insect food from the area discussed here, sub-Saharan Africa. You could present the situation of insects, especially their oils, as food source in some more details, probably backed up by references from the recent book volume on “African Edible Insects As Alternative Source of Food, Oil, Protein and Bioactive Components “, or similar resources

Response 4: Noted with thanks. The sections have been updated to capture the information requested. Additional references have been included to support the previous studies, see reference [9] in line 45 and other reference in lines 45 and 66.

Point 5. In the presentation, there are generally cross-relations between different components or parameters of cookies discussed, while occasionally some broader comparison would also be helpful.

Response 5: Thank you. The revised version compares values of omega 6 with other meat products, such as fish, beef and chicken, line 513.

Point 6. Fig. 9 shows that there are the strongest dislikes associated with products including insect oils – for all 5 parameters gaining a dislike ratio >50% of panelists come from insect oil products. For a costumer’s acceptance study, this should be spelled out at least in the discussion and could be considered even for the abstract, as it may hint at a fundamental difference in the acceptability of plant over insect oils. The one positive exception is the overall acceptability for the bakery product with Ruspolia oil. However, there may also be different types of food which could be better suited for insect oils? This is only vaguely hinted at in the conclusion (l. 601), but maybe you could elaborate on other uses for the insect oils.

Response 6: Point noted with thanks. We have included a statement in the abstract on the overall consumer acceptance of cookies prepared with insect oils (See line 23, 28-30). The higher preference for cookies baked with plant oils than insect oils is discussed in line 571-576 and in the text to highlight why such differences in consumer’s acceptability might occur. We have also added a statement as to why other better food sources should be explored for inclusion of insect oil other than bakery products (See line 576-580). About alternative use of insect oil, we have discussed that elaborately in the text (see Line 461- 607).

Point 7. For another example, you also carried out chemical analysis of olive and sesame oils de novo, and compare the composition for FAs in the discussion. The overall sum of FAs present in your insect and plant samples is stated (p. 279). But what about the similarities – just include a short mention if there are major differences, substances unique to one type of oil etc. What about the earlier studies of plant oils you cite – did they miss something, or did you not find a component in your sample reported previously? Would this possibly relate to the method of extraction?

Response 7: Thank you. We have included a statement to cover the fatty acids that were present across all the samples analyzed see lines 297-301. The discussion of oil results from other researchers is covered in line 480-494. Yes, the extraction procedure (line 461), state of sample (whether dry or wet) among other factors plays a key role in the observed variations

Point 8. The discussion should be structured better by breaking it down into sub-sections, e. g. 1. on the differences in oil chemistry between insects and plants, 2. Details on contents and their nutritional and health effects, and 3. the analysis of consumer acceptance results.

Response 8: Thank you. The discussion section has now been structured into two sections i.e. 4.1 Chemistry, nutritional and health properties of oils derived from insect and plant as well as their respective bakery product line 460 and 4.2 Sensory evaluation and consumer acceptability line 581

Point 9. Figures Figure 7 is far too small – is there something missing in the file provided for review?

Response 9: Figure 7 is now enhanced and made large

Point 10. There seems to be a mistake by exchanging the figures 8 and 10 – please revert! I’d also suggest to move figure 9 up in place of figure 8, to give an impression of the cookies first which are then discussed in the following.

Response 10. Thank you. The figures mix-up has now been corrected and the cookies figure brought first (Figure 8) followed by proportion of panelists preference for color, aroma, taste and overall acceptability of cookies fortified with insect-based oils (Figure 9) and The percentage of panelist willing to buy cookies prepared with insect-based oils (Figure 10).

Point 11. Tables.  Abbreviations are given at the end of the table. Since some tables are very long and extend over several pages, please move the abbreviations up so they are next to the heading and easily identify the meaning within the table on a first look.

Response 11: This has been corrected for Table 1 and Table 3

Point 12. l. 10 insect-derived ingredients

Response 12. A hyphen has been inserted between insect and derived (line 10)

Point 13. l. 12 knowledge… remains to be established.

Response 13. …’investigated’ has been replaced with ‘established’ (line 13)

Point 14. l. 20 What do you mean by proximate analysis? Are there alternatives?

Response 14. We wish to retain the term proximate analysis in line 20. This is because in food/feed analysis, proximate analysis entails partitioning of compounds in a food/feed into six categories based on the chemical properties of the compounds. The six categories are: Moisture, ash, crude protein (or Kjeldahl protein), crude lipid, crude fibre, nitrogen-free extracts (digestible carbohydrates)

Point 15. l. 21, 26 Consumer’s acceptance (it’s the acceptability of the cookie under scrutiny, not of the consumers)

Response 15. Consumer’s acceptability has now been changed to consumer’s acceptance see lines 21, 28, 75, 210, 244, 442, 445, 552, 561, 563, 575, 605

Point 16.l. 49 isn’t the emission of greenhouse gases and water and land requirement contributing to the ecological footprint?

Response 16. Yes, they do contribute to the ecological footprint. However, when insect rearing is compared to livestock/beef production, the former contribution of greenhouse gases, water and land requirement is minimal thus having a better ecological footprint. Line 52-55

Point 17.l. 53 – 57 Please add references for the numbers cited here.

Response 17. References of the report by ‘Persistence Market Research’ and “Grand View Research” have been provided. Line 60 [17] and line 62 [18]

Point 18.l. 100 delete comma

Response 18. The comma after Spain) has been deleted. In line 109

Point 19.l. 101 delete “and used for the study.”

Response 19. “and used for the study.” Has been deleted, line 110

Point 20. l. 212 It would be better if you keep with the parameters and their order given above (l. 200) to follow the analysis better, for example, the texture does not appear again later.

Response 20. Thank you, the parameter ‘texture’ has been deleted because it was not reported in this study. Line 221

Point 21. l. 435 …had a significantly higher crude protein content…

Response 21 the sentences has been rephrased accordingly, line 434

Point 22. l. 447 panelist’s acceptance

Response 22. Panelists’ acceptability has been changed to panelists’ acceptance line 442

Point 23. L. 470 Obviously the oil yield from insects depends on several extrinsic (diet, temperature) or intrinsic (development, age) factors. Another aspect could be differences in species for the fat/oil deposition in the body. As R. differens has a high body content of fat (J. N. Kinyuru et al. 2018, The Role of Edible Insects in Diets and Nutrition in East Africa), are there similar numbers for comparison available from S. gregaria?

Response 23. Yes, the value for S. gregaria is available and has been captured in the introduction section line 64-68. Also, we have added a statement in line 465 to ‘and method of oil extraction’ which is also a contributing factor to the oil yield.

Point 24. l. 508 …in living organisms including, in insects, it regulated… - something seems to be missing here in the argument – are two sentences fused?

Response 24. The sentence has been corrected line 497

Point 25. l. 548 “cultivar,” – delete free space

Response 25. Space after ‘cultivar’ has been deleted. Line 503

Point 26. The abbreviations used for FAs are given in the methods section (l. 220) and later in the results (l. 279-281) but FAMEs are only included on the methods (l. 104). After an extensive method description, it would be helpful to include also the abbreviation for FAMEs again (l. 293, 320).

Response 26. Thank you. FAMEs abbreviation has been described in line 305 and line 332

Reviewer 2 Report

Dear Authors,

Manuscript ID: foods-833203
Title: Chemistry and sensory characterization of bakery products prepared with oils from African edible insects
Reviewer comments

General

The study of oil of two edible insects is a relevant research topic.
The subject is of interest and may be relevant for Foods or journals involved in quality traits related composition and technology products.
A complete series of nutrients and compounds was analysed following the most innovative methodologies available in the food research.
The results may be of interest, but there are some data needs to be reviewed.

Title

It is informative and appropriate

Abstract

The overall presentation must be completed. The most important results should be reported; for example, in Lines 19-20 data can be added.
Line 25: the term “invisible” is inappropriate for the abstract and not suitable to a scientific paper concerning quality food where macro- and micronutrients were investigated. This sentence can be change considering another one in Conclusions.

Introduction

The state of the art is well documented and described. However, some references were missing.
-Line 53: the references of the report by “Persistence Market Research” must be given because data were detailed in relation to this paper.
-Line 56: the references of the report “Grand View Research” must be added in the literature.
Materials and methods
Line 94-: please add the quantities of the two insect oils were obtained in terms of yield.
Line 99-: please specify the quantities of the two oils were purchased to be carried out the study.
Line 187: please add the model of the pre-heated oven and the name of Company of Taranto city provided this apparatus.
Line 188: please add how many cookies were prepared
Line 191-: please report how many cookies were used per sample because quadruplicate samples is only mentioned in Line 197.

Results

The results of all the kinds of analyses are well reported by means of Figures and Tables. However, some aspects need to be more detailed.

Table 1: the legend of the measure unit of fatty acid composition, in terms of μg per mg of oil and cookies, is not clear. Please, at the first line of this Table, add the percentage of lipid content of insect oils (as in Figure 1) but also vegetable oils and cookies.

Table 4: please, specify that proximate analysis of cookies is expressed as it is. Add the lipid content of the different samples.

Discussion

Line 516-524: in this paragraph the Authors show the higher quantity of the linoleic acid (LA) of the samples compared to other foods of aquatic and terrestrial origin. Then, they report the beneficial effects on the humans’ health attributing to LA but in a wrong way.

Please, I recommend to consider in a different way the linoleic acid and the conjugated linoleic acid (CLA) because ONLY conjugated linoleic acid, a group of positional and stereoisomers of linoleic acid, has beneficial effects. In particular, two of them (c9t11-CLA and t10c12-CLA) have raised special attention as a result of their biological activity.

Please, separate the quantities of LA from the CLA and re-write the comment.

Author Response

Response to Reviewer 2 Comments

Point 1. General The study of oil of two edible insects is a relevant research topic. The subject is of interest and may be relevant for Foods or journals involved in quality traits related composition and technology products. A complete series of nutrients and compounds was analysed following the most innovative methodologies available in the food research. The results may be of interest, but there are some data needs to be reviewed.

Response 1. Thank you

Point 2. Title. It is informative and appropriate

Response 2.  Thank you. The title has been revised as per reviewer 1 suggestions to capture only one type of product tested, cookie. The title now reads ‘Chemistry and sensory characterization of a bakery product prepared with oils from African edible insects’

Point 3. Abstract. The overall presentation must be completed. The most important results should be reported; for example, in Lines 19-20 data can be added.

Response 3. A statement showing the of results of insect and plant oils is shown in line 18 and 19.

Point 4. Line 25: the term “invisible” is inappropriate for the abstract and not suitable to a scientific paper concerning quality food where macro- and micronutrients were investigated. This sentence can be change considering another one in Conclusions.

Response 4. Thank you. The term ‘invisible’ has been replaced with appropriate term from the conclusion section. line 26-28.

Point 5. Introduction The state of the art is well documented and described. However, some references were missing. -Line 53: the references of the report by “Persistence Market Research” must be given because data were detailed in relation to this paper.

Response 5. Reference of the report by ‘Persistence Market Research’ has been provide. Line 60 [17]

Point 6. -Line 56: the references of the report “Grand View Research” must be added in the literature.

Response 6. The reference of the report “Grand View Research” has been provided. Line 62 [18]

Point 7. Materials and methods. Line 94-: please add the quantities of the two insect oils were obtained in terms of yield.

Response 7. The oil yield calculated from the wet weight of insect is provided in line 105.

Point 8. Line 99-: please specify the quantities of the two oils were purchased to be carried out the study.

Response 8, The quantities of the two oils have been provided in line 109.

Point 9. Line 187: please add the model of the pre-heated oven and the name of Company of Taranto city provided this apparatus.

Response 9. The model is “BISTROT 665; BestFor®, Ferrara, Italy, has been added. Line 197

Point 10. Line 188: please add how many cookies were prepared

Response 10. Each treatment had a total of 850 cookies, therefore in total there were 4250 cookies. Line 197-198.

Point 11. Line 191-: please report how many cookies were used per sample because quadruplicate samples is only mentioned in Line 197.

Response 11. In each treatment, we had 850 cookies. This has been adjusted in the text. Line 197-198.

Point 12. Results. The results of all the kinds of analyses are well reported by means of Figures and Tables. However, some aspects need to be more detailed. Table 1: the legend of the measure unit of fatty acid composition, in terms of μg per mg of oil and cookies, is not clear. Please, at the first line of this Table, add the percentage of lipid content of insect oils (as in Figure 1) but also vegetable oils and cookies.

Response 12. Thank you for the suggestion. Unfortunately, we did not measure lipid content of the oils and the cookies as well as that of the vegetable oils and their respective products. We have taken note of this to include such information in future studies

Point 13. Table 4: please, specify that proximate analysis of cookies is expressed as it is. Add the lipid content of the different samples.

Response 13. Thank you. We have added the phrase ‘specify that proximate analysis of cookies is expressed as it is’ as a foot note for Table 4.

Point 14. Discussion: Line 516-524: in this paragraph the Authors show the higher quantity of the linoleic acid (LA) of the samples compared to other foods of aquatic and terrestrial origin. Then, they report the beneficial effects on the humans’ health attributing to LA but in a wrong way.

Response 14. Thank you. The role of linoleic acid in humans is now correctly captured in line 513-515. ‘……Linoleic acid is widely known for its role in synthesis of arachidonic acid a precursor for various hormones, such as prostaglandins, thromboxanes, and leukotrienes used for the regulations of many physiological process’ and a reference provided for the same [54]

Point 15. Please, I recommend to consider in a different way the linoleic acid and the conjugated linoleic acid (CLA) because ONLY conjugated linoleic acid, a group of positional and stereoisomers of linoleic acid, has beneficial effects. In particular, two of them (c9t11-CLA and t10c12-CLA) have raised special attention as a result of their biological activity. Please, separate the quantities of LA from the CLA and re-write the comment.

Response 15. Thank you.  We have discussed the two fatty acids linoleic acid and α-linolenic acid (Z, Z, Z)-9,12,15-octadecatrienoate (ALA) together because the two are classified as essential fatty acids. We grouped the results together based on the statistics analysis (line 350-357 and line 327) and only discussed the results of linoleic acid and α-linolenic acid as they are among beneficial fatty acids with great biological activities.

Reviewer 3 Report

Very interesting study with relevant results. However, the authors should address, from an economic point of view, how profitable it is to produce these insect-based foods on a large scale. To what extent can it compensate for the food industry and will have a high level of consumers when compared with other foods.

Some important and recent studies should be discussed:

Di Mattia, C., Battista, N., Sacchetti, G., & Serafini, M. (2019). Antioxidant activities in vitro of water and liposoluble extracts obtained by different species of edible insects. Frontiers in Nutrition6, 106.

Elhassan, M., Wendin, K., Olsson, V., & Langton, M. (2019). Quality aspects of insects as food—nutritional, sensory, and related concepts. Foods8(3), 95.

Idowu, A. B., Oliyide, E. O., Ademolu, K. O., & Bamidele, J. A. (2019). Nutritional and anti-nutritional evaluation of three edible insects consumed by the Abeokuta community in Nigeria. International Journal of Tropical Insect Science39(2), 157-163.

Kauppi, S. M., Pettersen, I. N., & Boks, C. (2019). Consumer acceptance of edible insects and design interventions as adoption strategy. International Journal of Food Design4(1), 39-62.

Köhler, R., Kariuki, L., Lambert, C., & Biesalski, H. K. (2019). Protein, amino acid and mineral composition of some edible insects from Thailand. Journal of Asia-Pacific Entomology22(1), 372-378.

Manditsera, F. A., Luning, P. A., Fogliano, V., & Lakemond, C. M. (2019). The contribution of wild harvested edible insects (Eulepida mashona and Henicus whellani) to nutrition security in Zimbabwe. Journal of Food Composition and Analysis75, 17-25.

Montowska, M., Kowalczewski, P. Ł., Rybicka, I., & Fornal, E. (2019). Nutritional value, protein and peptide composition of edible cricket powders. Food Chemistry289, 130-138.

Patel, S., Suleria, H. A. R., & Rauf, A. (2019). Edible insects as innovative foods: Nutritional and functional assessments. Trends in Food Science & Technology.

Raheem, D., Carrascosa, C., Oluwole, O. B., Nieuwland, M., Saraiva, A., Millán, R., & Raposo, A. (2019). Traditional consumption of and rearing edible insects in Africa, Asia and Europe. Critical reviews in food science and nutrition59(14), 2169-2188.

Raheem, D., Raposo, A., Oluwole, O. B., Nieuwland, M., Saraiva, A., & Carrascosa, C. (2019). Entomophagy: Nutritional, ecological, safety and legislation aspects. Food Research International, 108672.

Sogari, G., Amato, M., Biasato, I., Chiesa, S., & Gasco, L. (2019). The Potential role of insects as feed: A Multi-Perspective Review. Animals9(4), 119.

Author Response

Response to Reviewer 3 Comments

Point 1. Very interesting study with relevant results. However, the authors should address, from an economic point of view, how profitable it is to produce these insect-based foods on a large scale. To what extent can it compensate for the food industry and will have a high level of consumers when compared with other foods.

Response 1.  The first advantage is that the method of oil extraction from the insects is very simple and cost-effective, simply using aqueous media and high-level centrifugation. (Line 76-83) This can easily be employed at large scale industrial level. Also, insects can significantly compensate for the food industries as novel food because insects are possible alternative to animal protein source thanks to their richness in protein, fat, minerals and vitamins (Collins et al. 2019), lower request of land and water (Van Huis 2013), lower environmental impacts in terms of fewer greenhouse gases emissions and ammonia production (Oonincx et al. 2010), and also due to their more efficient feed conversion rate with respect to conventional meats (Schlup and Brunner 2018; Mancini et al. 2019). Insect farming is a very low-cost technology with low capital investment and labor involved. Insects can also be an excellent source of regular cash flow due to their very short generation time and high reproductive rate, thus affordable for everyone interested in participating in the value chain. Line 50-54.

Point 2. Some important and recent studies should be discussed. Di Mattia, C., Battista, N., Sacchetti, G., & Serafini, M. (2019). Antioxidant activities in vitro of water and liposoluble extracts obtained by different species of edible insects. Frontiers in Nutrition6, 106.

Response 3. This reference has been discussed in line 475-476, reference [39]

Point 3. Elhassan, M., Wendin, K., Olsson, V., & Langton, M. (2019). Quality aspects of insects as food—nutritional, sensory, and related concepts. Foods8(3), 95.

Response 3. This reference has been discussed in line 590-591, reference [76]

Point 4. Idowu, A. B., Oliyide, E. O., Ademolu, K. O., & Bamidele, J. A. (2019). Nutritional and anti-nutritional evaluation of three edible insects consumed by the Abeokuta community in Nigeria. International Journal of Tropical Insect Science39(2), 157-163.

Response 4. Thank you. This reference has been discussed in line 549 [64]

Point 5. Kauppi, S. M., Pettersen, I. N., & Boks, C. (2019). Consumer acceptance of edible insects and design interventions as adoption strategy. International Journal of Food Design4(1), 39-62.

Response 5. This reference has been discussed in line 581, reference [76]

Point 6. Köhler, R., Kariuki, L., Lambert, C., & Biesalski, H. K. (2019). Protein, amino acid and mineral composition of some edible insects from Thailand. Journal of Asia-Pacific Entomology22(1), 372-378.

Response 6. This reference has been discussed in line 549 reference [63]

Point 7. Manditsera, F. A., Luning, P. A., Fogliano, V., & Lakemond, C. M. (2019). The contribution of wild harvested edible insects (Eulepida mashona and Henicus whellani) to nutrition security in Zimbabwe. Journal of Food Composition and Analysis75, 17-25.

Response 7. Thank you. The reference has been discussed in line 47, reference [10].

Point 8. Montowska, M., Kowalczewski, P. Ł., Rybicka, I., & Fornal, E. (2019). Nutritional value, protein and peptide composition of edible cricket powders. Food Chemistry289, 130-138.

Response 8. This reference has been discussed in line 52, reference [15].

Point 9. Patel, S., Suleria, H. A. R., & Rauf, A. (2019). Edible insects as innovative foods: Nutritional and functional assessments. Trends in Food Science & Technology.

Response 9. This reference has been discussed in line 39, reference [3].

Point 10. Raheem, D., Carrascosa, C., Oluwole, O. B., Nieuwland, M., Saraiva, A., Millán, R., & Raposo, A. (2019). Traditional consumption of and rearing edible insects in Africa, Asia and Europe. Critical reviews in food science and nutrition59(14), 2169-2188.

Response 10. This reference has been discussed in line 47, reference [11]

Point 11. Raheem, D., Raposo, A., Oluwole, O. B., Nieuwland, M., Saraiva, A., & Carrascosa, C. (2019). Entomophagy: Nutritional, ecological, safety and legislation aspects. Food Research International, 108672.

Response 11. This reference has been discussed in line 591, reference [74]

Point 12. Sogari, G., Amato, M., Biasato, I., Chiesa, S., & Gasco, L. (2019). The Potential role of insects as feed: A Multi-Perspective Review. Animals9(4), 119.

Response 12. This reference has been discussed in line 591, reference [75]

Reviewer 4 Report

The manuscript from Cheseto and colleagues reports on the characterization of chemistry profiles of the oil extracted from two commonly consumed grasshoppers in sub-Saharan Africa and analysis of the nutrients and flavors of foods prepared by using these oils. Overall, this is a simple but interesting study with clear purpose of research and the experiments seem to have been carried out in a sound manner. There are some minor issues should be considered, which are listed below.

  1. Line 43, “icipe”, the full name of the organization should be provided here.
  2. Line 52-55, reference needed for this sentence.
  3. Line 56-58, reference needed for this sentence.
  4. “P” for P-values has to be in consistently italic throughout the text.
  5. Table 4, foot notes are missing.

Author Response

Response to Reviewer 4 Comments

Point 1. The manuscript from Cheseto and colleagues reports on the characterization of chemistry profiles of the oil extracted from two commonly consumed grasshoppers in sub-Saharan Africa and analysis of the nutrients and flavors of foods prepared by using these oils. Overall, this is a simple but interesting study with clear purpose of research and the experiments seem to have been carried out in a sound manner. There are some minor issues should be considered, which are listed below.

Response 1.  Thank you

Point 2. Line 43, “icipe”, the full name of the organization should be provided here.

Response 2. The full name of the organization International Centre of Insect Physiology and Ecology has been provided. Line 47-48

Point 3. Line 52-55, reference needed for this sentence.

Response 3. The refence is now provided for line 60. Reference [17]

Point 4. Line 56-58, reference needed for this sentence.

Response 4. The refence is now provided for line 62. Reference [18]

Point 5. “P” for P-values has to be in consistently italic throughout the text.

Response 5. P-values has been italicized and capitalized throughout the text. line 226, 250, and 380 – 388

Point 6. Table 4, foot notes are missing.

Response 6. Thank you. We have added the foot note for Table 4

Round 2

Reviewer 3 Report

In my opinion, now the manuscript is suitable for publication in Foods.